# Combining machine learning with high-content imaging to infer ciprofloxacin susceptibility in isolates of *Salmonella* Typhimurium

Tuan-Anh Tran [1,2,3,18], Sushmita Sridhar [1,4,18], Stephen T. Reece[1,5], Octavie Lunguya[6,7], Jan Jacobs [8,9], Sandra Van Puyvelde [1,10], Florian Marks [1,11,12,13], Gordon Dougan[1], Nicholas R. Thomson[4,14], Binh T. Nguyen[15], Pham The Bao[16] & Stephen Baker [1,17] ✉

Antimicrobial resistance (AMR) is a growing public health crisis that requires innovative solutions. Current susceptibility testing approaches limit our ability to rapidly distinguish between antimicrobial-susceptible and -resistant organisms. *Salmonella* Typhimurium (*S.* Typhimurium) is an enteric pathogen responsible for severe gastrointestinal illness and invasive disease. Despite widespread resistance, ciprofloxacin remains a common treatment for *Salmonella* infections, particularly in lower-resource settings, where the drug is given empirically. Here, we exploit high-content imaging to generate deep phenotyping of *S.* Typhimurium isolates longitudinally exposed to increasing concentrations of ciprofloxacin. We apply machine learning algorithms to the imaging data and demonstrate that individual isolates display distinct growth and morphological characteristics that cluster by time point and susceptibility to ciprofloxacin, which occur independently of ciprofloxacin exposure. Using a further set of *S.* Typhimurium clinical isolates, we find that machine learning classifiers can accurately predict ciprofloxacin susceptibility without exposure to it or any prior knowledge of resistance phenotype. These results demonstrate the principle of using high-content imaging with machine learning algorithms to predict drug susceptibility of clinical bacterial isolates. This technique may be an important tool in understanding the morphological impact of antimicrobials on the bacterial cell to identify drugs with new modes of action.

Antimicrobial resistance (AMR) is a mounting global health issue, which on a patient-by-patient basis, narrows the therapeutic options for selecting appropriate antimicrobial agents[1,2]. There have been significant advances in understanding AMR mechanisms in bacteria, including the differential roles of chromosomal mutations, plasmid-borne resistance genes, and inducible resistance or hetero-resistance to drug exposure[3–5]. However, this improved understanding has not increased the ability to rapidly distinguish antimicrobial susceptible and resistant organisms. Conventional phenotypic antimicrobial susceptibility testing (AST) is a multi-day process that relies on isolation of

a single bacterial colony, growth of a bacterial suspension, followed by incubation of the culture with an antimicrobial and reading and interpretation by a laboratory technician or by machine in well-resourced settings[6,7]. Automated AST methods such as the Vitek2 and BD Phoenix increase throughput and decrease turnaround time but still rely on conventional readouts of bacterial growth in the presence of antimicrobial[8]. This modality for AST results often results in patients being treated empirically with an inappropriate antimicrobial, with negative implications for patient outcomes[9–11]. Thus, we still require adaptable approaches that can rapidly discriminate between susceptible and resistant organisms without the requirement of classical phenotypic AST.

High-content imaging (HCI) integrates automated high-resolution microscopy and analysis pipelines to measure a multitude of morphological variables in individual cells within a defined population[12]. Therefore, HCI can reproducibly capture the characteristics and heterogeneity within a microscopic population without compromising measurement representativeness. The most common application of HCI in bacteriology is to measure morphological changes under exposure to drugs with a known mode of action (MoA)[13–18]. This approach facilitates the qualitative prediction of the MoA of new compounds based on their dimensional characteristics. Such techniques can be applied to study how bacterial populations respond to antimicrobials. Therefore, the subtle scrutiny of standardized morphological features may predict how AMR arises and how an organism will respond to any given chemical perturbation. Studies have begun to investigate these links using HCI at bacterial single-cell resolution paired with image analysis, but the relationship between such morphological characteristics and AMR is not yet well defined[13,19,20]. Given the assumption that bacterial populations do not behave uniformly, characterizing any variation within and between large bacteria populations requires complex analytical methods beyond the scope of basic statistical approaches.

*Salmonella enterica subsp. enterica* serovar Typhimurium (*S.* Typhimurium), a classical enteric pathogen, which can induce gastroenteritis, diarrhea, and in some cases, systemic disease[21–24]. Pertinently, *S.* Typhimurium is becoming increasingly resistant to key antimicrobials, including the broad-spectrum fluoroquinolone, ciprofloxacin, which acts by stalling replication via permanent double-stranded DNA breaks[25–27]. Ciprofloxacin remains a key antimicrobial against invasive *Salmonella* infections, and thus, there is a need to better characterize *Salmonella* resistance to ciprofloxacin at the cellular level[28,29].

Aiming to better understand the relationship between cellular morphology and AMR, we sought to link HCI with machine learning algorithms to identify the key characteristics that may predict how a bacterial population responds to an antimicrobial. To address this problem, we selected two laboratory-typed strains and two clinically relevant *S.* Typhimurium isolates, expose them to four concentrations of ciprofloxacin over a 24-hour timeframe, and periodically perform HCI[21–27]. Using detailed machine learning algorithms on the imaging data, we show that *S.* Typhimurium have distinct morphological characteristics that can be exploited to predict ciprofloxacin susceptibility without prior knowledge of susceptibility phenotype or exposure to the antimicrobial.

## Results

### S. Typhimurium isolates display variable growth and morphological characteristics following ciprofloxacin exposure

We selected two clinical isolates and two isogenic laboratory strains of *S.* Typhimurium with a range of ciprofloxacin susceptibilities (D23580, 0.03 µg/ml; VNS20081, 1.0 µg/ml; SL1344, 0.015 µg/ml; and SL1344*gyrA*, 1.5 µg/ml) to subject to sustained ciprofloxacin exposure (Table 1). Both VNS20081 and SL1344*gyrA* have a mutation within the quinolone-resistance determining region (QRDR) of GyrA, a subunit of

**Table 1 | Isolates, ciprofloxacin susceptibility, and in silico analysis of genetic mutations related to fluoroquinolone susceptibility**

| Isolate | Ciprofloxacin (S,I,R) | Ciprofloxacin MIC (µg/ml) | GyrA mutation in QRDR* | GyrA mutation outside QRDR | GyrB mutation in QRDR | qnr | acrA/acrB | oqxA/oqxB |
|---|---|---|---|---|---|---|---|---|
| SL1344 | S | 0.015 | no | no | no | no | no | no |
| SL1344gyrA | R | 1.5 | yes (D87Y) | yes (A352S, D200N, E283D, V250A, V385I) | no | no | no | no |
| D23580 | S | 0.03 | no | yes (D200N, V250A) | no | no | no | no |
| VNS20081 | R | 1.0 | yes (D87N) | yes (D200N) | no | no | no | yes |
| VNB1779 | I | 0.75 | yes (D87N) | no | no | no | yes (acrB) | yes |
| VNB2315 | S | 0.03 | no | no | no | no | yes (acrB) | no |
| gha113289 | I | 0.75 | no | yes (A352S, D200N, E283D, V250A, V385I) | no | no | yes (acrB) | no |
| gha200597 | S | 0.016 | no | yes (D200N) | no | no | no | no |
| 2101 | S | 0.032 | no | yes (D483E, L447Q, R490H) | no | no | yes (acrB) | no |
| 16755_3 | I | 0.375 | yes (S83Y) | no | no | no | no | no |
| 5390_4 | I | 0.5 | no | yes (D200N, D483E, L447Q, L506M, R490H, V250A) | no | yes (qnrS) | no | no |
| 8314_12 | I | 0.19 | no | yes (D200N, D483E, L447Q, R490H, V250A) | yes (S464F) | no | no | no |
| 8599_13 | S | 0.023 | no | yes (D200N, V250A) | no | no | no | no |
| 319_8 | I | 0.25 | yes (D87Y) | yes (D200N, E283D, V250A) | no | no | no | no |
| 1304 | S | 0.023 | no | yes (D200N, V250A) | no | no | no | no |
| D23580gyrA | I | 0.5 | yes (D87G) | yes (D200N, F145S, F96V, G170S, G71D, T142P, T188M) | no | no | yes (acrA) | no |
| 10433_3 | S | 0.03 | no | yes (D200N, E283D, V250A) | no | no | yes (acrB) | no |

Isolates were selected based on ciprofloxacin susceptibility phenotype. Ciprofloxacin susceptibility was assessed using ciprofloxacin test strips using CLSI guidelines to distinguish isolates as sensitive (S), intermediate (I), or resistant (R). In silico AMR analysis was conducted with ARIBA using the Comprehensive Antimicrobial Resistance Database (CARD) on reads files and with the Resistance Gene Identifier, which also uses CARD, on assemblies. *QRDR quinolone resistance determining region.

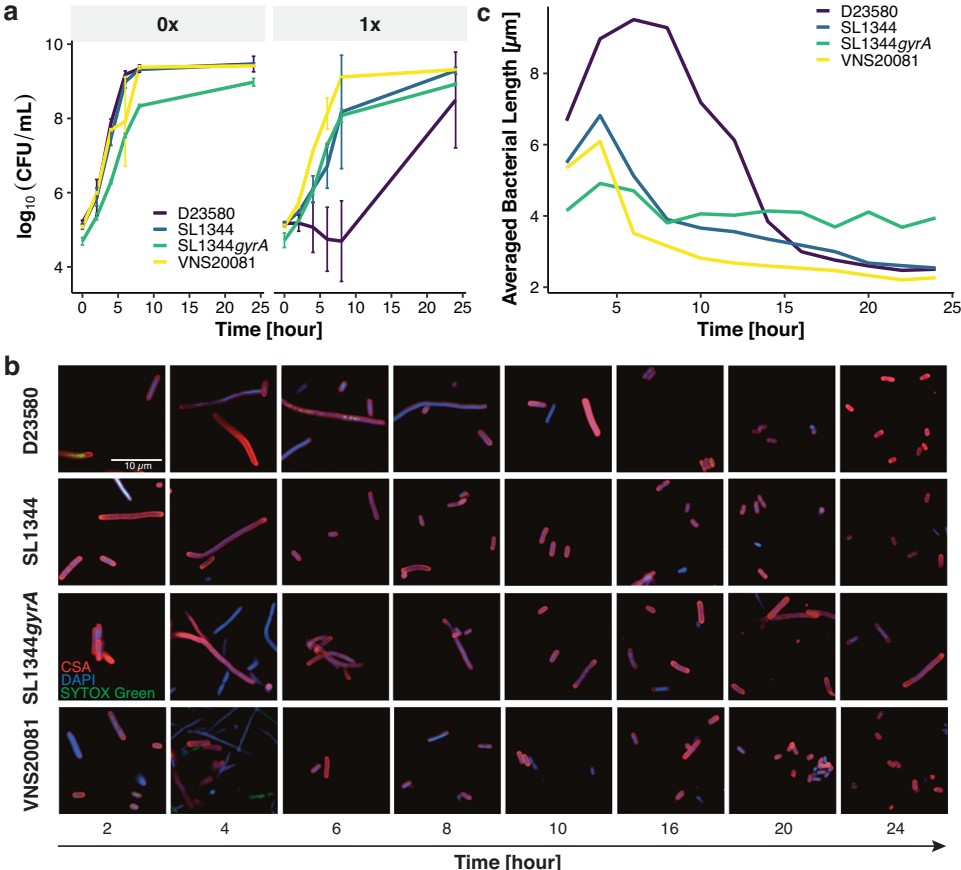

**Fig. 1 | *S*. Typhimurium responses to ciprofloxacin treatment. a** Time-killing curves of the four isolates (*n* = 3 biological replicates) at 0xMIC (left subplot) and 1xMIC (right subplot) against exposure time (horizontal axis). Vertical axis is the average of colony forming units in logarithm scale. Vertical bars represent standard deviation. **b** Morphological responses of *S*. Typhimurium isolates at 1xMIC against increasing exposure time (from left to right). CSA, SYTOX Green, and DAPI stains give red, green, and blue fluorescence, respectively. Representative images were chosen from one well of one experiment of three biological replicates. **c** Averaged length over all bacterial individuals at 1xMIC. Horizontal axis is exposure time in hours. Vertical axis is averaged length in micrometers. Dark purple line is D23580, blue line is SL1344, green line is SL1344*gyrA*, and yellow line is VNS20081. Source data are provided as a Source Data file.

DNA gyrase. A mutation within the QRDR of GyrA is the primary mechanism of fluoroquinolone resistance in Gram-negative bacteria[30,31]. We performed time kill curves with the MIC of ciprofloxacin, referred to as "1 × MIC", and without antimicrobial as a baseline comparator, enumerating colony-forming units (CFU) every two hours for eight hours and after 24 h (Fig. 1a).

All four strains displayed comparable growth dynamics in the absence of ciprofloxacin, but we observed greater variation in the growth trajectories between strains at 1 × MIC of ciprofloxacin. Overall, growth was slower in the presence of ciprofloxacin, with D23580 (dark purple line) showing the greatest reduction in growth rate compared to other strains and to D23580 without the drug (Fig. 1a). Despite a reduced growth rate over the initial 8 h of ciprofloxacin exposure, all strains treated with 1 × MIC ciprofloxacin rebounded to a CFU comparable to the no ciprofloxacin control by the 24 h time point. These dynamics show that the four bacterial strains with their different genetic compositions exhibit distinct growth trajectories in the presence of ciprofloxacin. We postulated that the observed differences may be measured more precisely by assessing changes in cellular morphology and detailed phenotypes; therefore, we subjected the strains to HCI to identify morphological differences associated with ciprofloxacin exposure.

The four *S*. Typhimurium strains were again grown aerobically in liquid media at 0 × , 1 × , 2 × , and 4 × MIC ciprofloxacin for 24 h; samples were taken every two hours for imaging. Cells were treated with three stains to capture multiple cellular compartments and features:

CSA (bacterial membrane), DAPI (nucleic acids), and SYTOX Green ("SG") (dead/damaged cell)[32]. Over the course of the experiment, we observed a temporal change in bacterial length under 1 × MIC ciprofloxacin treatment for up to 24 h, due to the cell's inability to septate, as has been observed in previous studies[13,15,16,18,33] (Fig. 1b). This change in bacterial cell length was more pronounced between 2 and 8 h, although strain SL1344*gyrA* (containing a GyrA mutation that confers resistance to ciprofloxacin) could be distinguished by sustained cellular elongation at 24 h (Fig. 1b).

Our HCI platform captured 65 morphological, intensity, and texture features for each individual bacterial cell within the selected field-of-view[20], generating the power to quantify mean bacterial cell length for an entire field at each time point. At 1 × MIC ciprofloxacin, the mean cellular length peaked at 9.51 ± 0.20 μm (mean ± standard error) within six hours, which was a predicted consequence of fluoroquinolone exposure[34], and subsequently decreased below the initial length (Fig. 1c). We observed several strain-specific disparities: D23580 exhibited greater cell elongation within six hours, and strain SL1344*gyrA* exhibited only a modest decrease in mean cellular length after eight hours of exposure to 1 × MIC of ciprofloxacin. Notably, D23580 and VNS20081 grown with ciprofloxacin supplementation displayed considerable heterogeneity in cell length over the course of the experiment, with the degree of cell length heterogeneity being dependent on the strain, the ciprofloxacin concentration, and also the exposure time (Fig. S1). Overall, the general trend across all isolates at 1 × MIC of ciprofloxacin was CFU counts by 24 h comparable to no

ciprofloxacin, a significant increase in bacterial length with time followed by a decrease, which was quantified and visually prominent.

## Development of a random forest classifier to discriminate between isolates and between growth conditions

We next sought to disaggregate the role of bacterial strain from treatment condition. For every strain, the ciprofloxacin exposure, morphology, intensity, and texture features were averaged across individual bacteria on a single focal plane within a microwell. The averaged data was then collated as one datapoint, hereafter referred to as imaging data/features. Subsequently, we normalized the imaging data to z-scores, which was visualized on a heatmap plot with hierarchical clustering (Fig. 2a). The resulting data was segregated into two main groups, with the majority of the 0 × MIC and 1 × MIC imaging datapoints in one major cluster and the 2 × MIC and 4 × MIC datapoints in a second cluster (Fig. 2a). To compare the contribution of time to patterns of the data, these data were projected into two-dimensional space using principal coordinate analysis (PCoA) (Fig. 2b). The two principal axes explained 98.6% variation of the data, and we observed a time-associated left-to-right (purple to yellow) pattern for the four strains with increasing exposure time to ciprofloxacin. These heatmap and PCoA clustering suggested that, of the factors measured, drug concentration and exposure time had the greatest effect on the cellular response to ciprofloxacin treatment.

We next sought to identify imaging features associated with the drug concentration and exposure time. We trained a random forest classifier on the imaging data from merged data of all strains to classify the treatment conditions, which were a combination of exposure time and ciprofloxacin concentrations (i.e., 0 × −2h, 0 × −4h,..., 4 × −24h, etc). The random forest yielded an acceptable predictive performance with an out-of-bag (OOB) error rate of 0.25, signifying that this model would allow us to effectively identify important imaging features in our data (Fig. 2c). The ten most important features for the combined strain dataset, corresponding to the highest calculated prominence, were determined by the random forest classifier (Fig. 2d). The importance score (x-axis) calculated for each feature displays how much the identified feature contributes to the predictive performance of a machine learning model, most notably the random forest model in this study. Seven of these features were associated with fluorescence intensity and texture (SG intensity mean, SG intensity SD, SG relative radial deviation, SG profile 2.2, SG profile 1.2, CSA radial mean, and DAPI compactness); three features were associated with bacterial morphology (bacterial cell area, spot area (a size boundary used to identify bacteria), and bacterial cell length).

We next plotted the mean bacterial cell length at different ciprofloxacin concentrations and exposure time to build upon our earlier analysis of mean bacterial length at 1 × MIC of ciprofloxacin (Fig. 1c and S2a). These results exhibited a time-dependent trend for bacterial cell length. Specifically, strains D23580, VNS20081, and SL1344 responded to ciprofloxacin exposure in a biphasic manner: (1) bacterial cells initially elongated, and then (2) reduced in length with time. In contrast, the bacterial length trend for the SL1344*gyrA* mutant was uniform. These trends in bacterial cell length were comparable for the conditions of 1 × , 2 × , and 4 × ciprofloxacin MIC. We observed a linear correlation ($\rho = 0.72$) between drug concentration and the time for isolates D23580, SL1344, and VNS20081 to reach its peak average length (Fig. 2e). This finding demonstrated how the interrelatedness of exposure time and concentration impacts bacterial length. We additionally observed that bacterial cell length became more similar over time for individual cells belonging to strains D23580, VNS20081, and SL1344, suggesting that there were more morphological differences within and between *S.* Typhimurium strains treated with different ciprofloxacin concentrations earlier during ciprofloxacin exposure (Fig. S2a). Images of D23580 and VNS20081 from discrete time points and across all ciprofloxacin concentrations indicated in most cases a

convergence of bacterial cell length, although this was the most pronounced in VNS20081, and D23580 at 4× MIC from 18 h was a notable exception (Fig. S2b, c). Overall, we could train a random forest classifier to identify important features common across bacterial strains, and, using cell length as an exemplar of an important feature, we were able to examine how cell length changed over time and differed between concentrations of ciprofloxacin.

## S. Typhimurium SL1344 and its GyrA mutant display distinct differences in imaging features

While it was important to recognize that these four strains of *S.* Typhimurium had distinct morphological features, we additionally wanted to investigate whether SL1344 and its ciprofloxacin-resistant GyrA mutant derivative exhibited observable differences in cellular morphology. To identify the treatment condition that segregated SL1344 and SL1344*gyrA* with the highest degree of distinction, we plotted their distance ratio of the four ciprofloxacin concentrations against exposure time. The distance ratio was minimized at 4 × MIC and 20 h (4 × MIC-20h), corresponding to the most separation between the two isolates (Fig. 3a). Therefore, subsequent analyses were conducted at 4 × MIC at the 20 h time point. We found that SL1344 and SL1344*gyrA* segregated well at 4 × MIC-20h, with two independent 95%-confidence ellipses (Fig. 3b). A random forest classifier was trained to classify the two organisms, yielding an OOB error rate of 0, equivalent to 100% predictive performance. We then derived the ten most important features from the trained random forest classifier and plotted them against the corresponding highest calculated importance, shown for SL1344 and SL1344*gyrA* combined, and for each strain (Fig. 3c). We plotted the ten most important features of SL1344 and SL1344*gyrA* individually, finding that each pairwise comparison was highly significant ($p = 0.00075$) (Fig. S3). A comparison of representative images from 4 × MIC-20h showed that SL1344 and SL1344*gyrA* were distinct (Fig. 3e).

These ten important features were combined into a radar chart as observable differences between organisms (Fig. 3d). These data indicated that while SL1344 and SL1344*gyrA* are isogenic, the GyrA mutation resulted in substantial morphological differences when the bacteria were exposed to ciprofloxacin. More broadly, these data suggest that a mutation in the quinolone resistance determining region (QRDR) of GyrA may have further global effects in addition to reduced ciprofloxacin susceptibility, and the ability to identify important morphological features for a given isolate, may provide novel insights for inferring ciprofloxacin susceptibility.

## Ciprofloxacin susceptible and resistant S. Typhimurium isolates can be distinguished by imaging features

Having quantitatively distinguished *S.* Typhimurium SL1344 from a ciprofloxacin resistant SL1344*gyrA* mutant strain, we sought to discriminate between ciprofloxacin susceptible and resistant isolates with diverse genetic backgrounds. Notably, ciprofloxacin resistant *S.* Typhimurium exhibited the most distinction from susceptible isolates after 22 h of culture without exposure to ciprofloxacin, which was quantified by the lowest distance ratio at the condition of 0 × MIC-22h (Fig. 4a). This observation suggests that DNA gyrase mutations may interact with other cellular pathways, which ultimately impact directly on cellular morphology.

We subsequently investigated how susceptibility to ciprofloxacin affects the distribution of imaging data at 0 × MIC-22h. Projecting all 65 imaging features at 0 × MIC-22h onto a PCoA plot (explaining 98.6% variation), we found that the resistant organisms clearly segregated from susceptible organisms (Fig. 4b). We next trained a random forest model to select the ten most important features associated with ciprofloxacin resistance at 0 × MIC-22h for further analysis. The random forest model possessed a high predictive performance at 0 × MIC-22h in the best segregation of ciprofloxacin susceptible and resistant

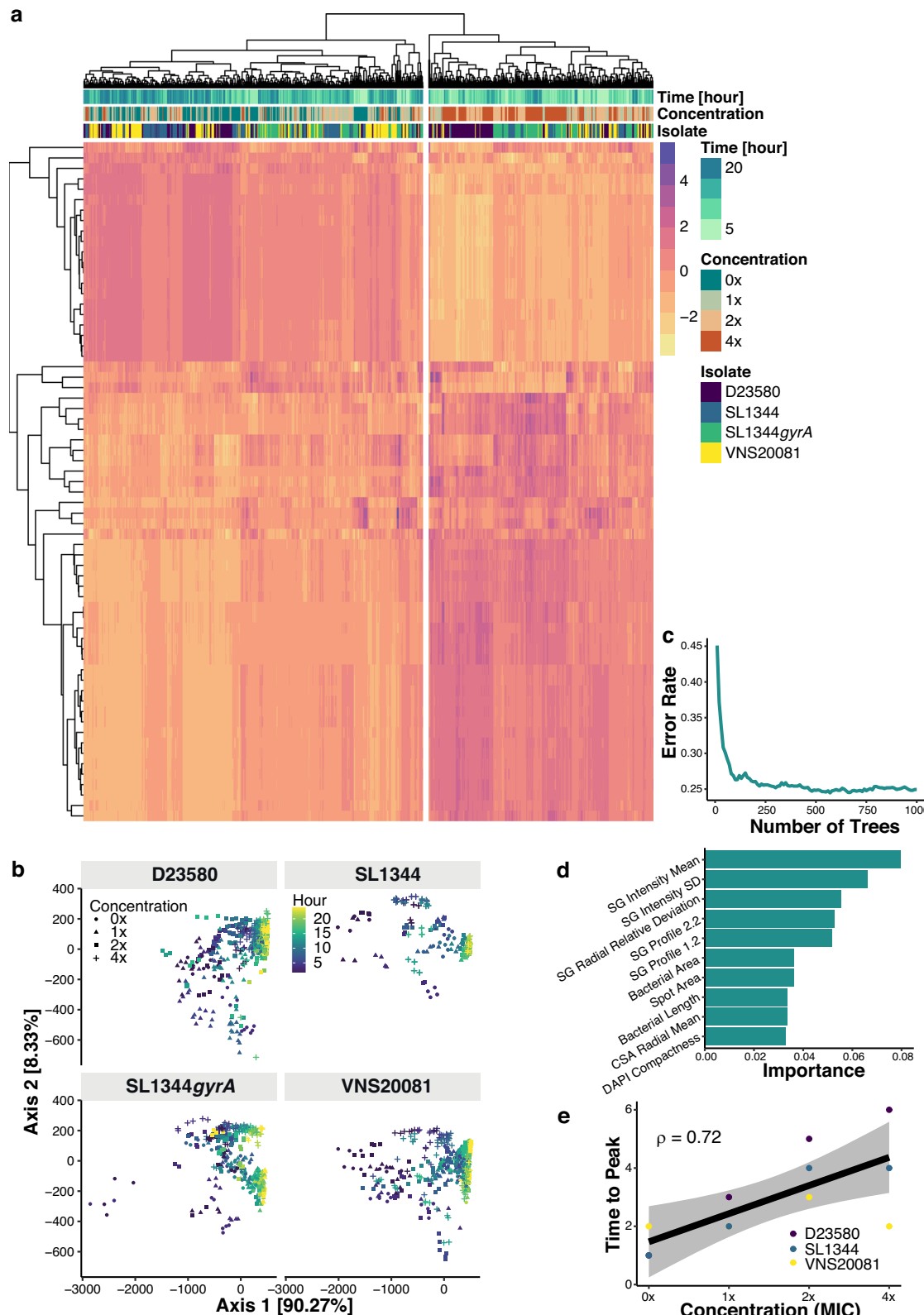

**Fig. 2 | Morphological features associated with ciprofloxacin exposure.**
**a** Feature heatmap with hierarchical clustering. Horizontal axis is datapoints, vertical axis is features, and each cell in the heatmap is the z-score value of a feature of a datapoint. **b** Two-dimensional scatter PCoA plot of *S*. Typhimurium treated with ciprofloxacin at 0xMIC (circle), 1xMIC (triangle), 2xMIC (square) and 4xMIC (cross). Colour scale represents time of exposure. **c** Random forest OOB error rate against number of decision tree used in ensembles. **d** Feature importance extracted from random forest model with 1000 decision trees to distinguish different treatment conditions. The horizontal axis is importance, and the vertical axis is the ten most important features. SYTOX Green is abbreviated as SG. **e** Correlation between ciprofloxacin concentration relative to MICs (horizontal axis) and required time for each isolate to elongate to maximum length (vertical axis). The gray band represents 95%-confidence interval for the linear regression. Source data are provided as a Source Data file.

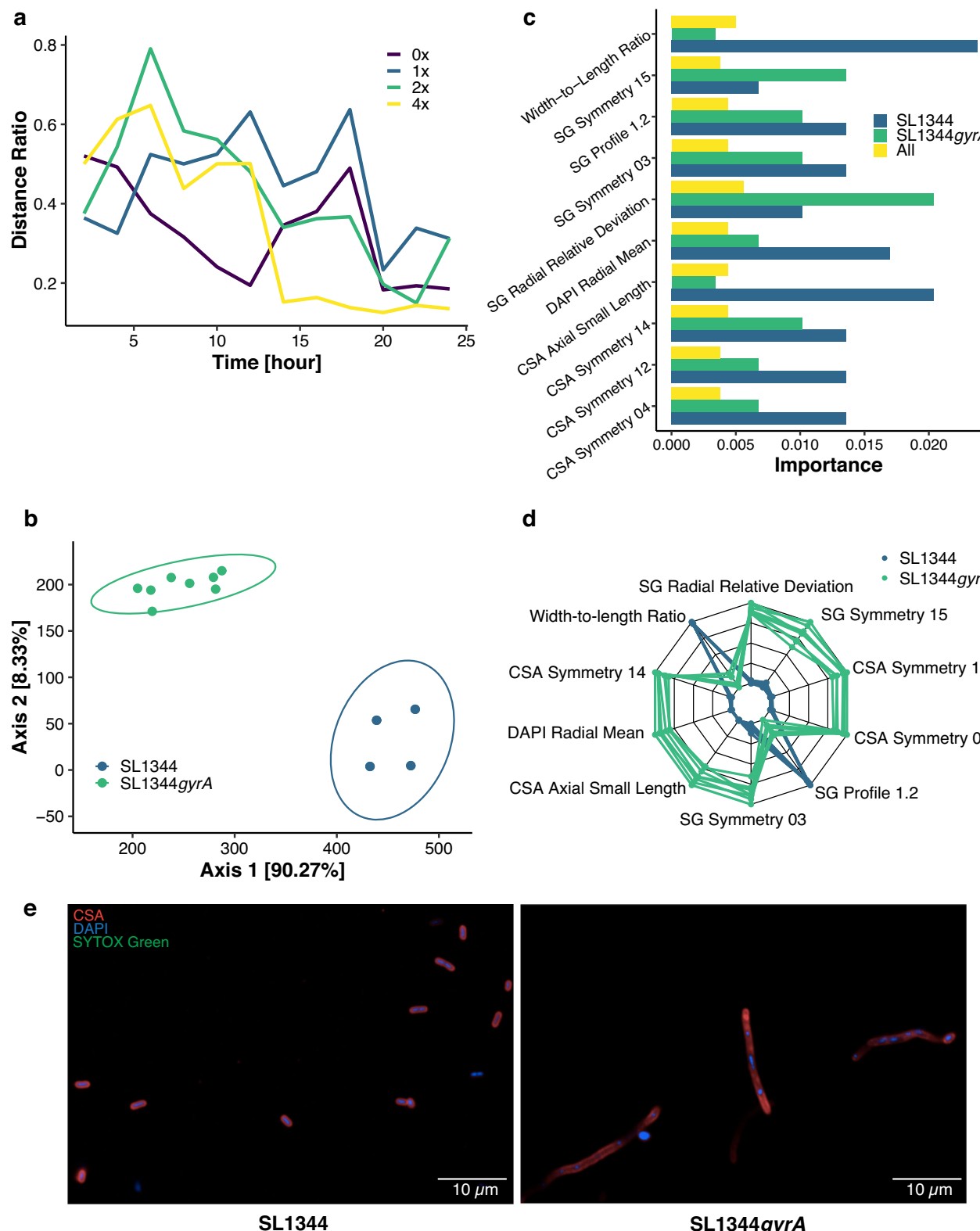

**Fig. 3 | Morphological distinction between SL1344 and SL1344*gyrA*. a** Distance ratio, between datapoints grouped by the isolate SL1344 and its mutant derivate SL1344*gyrA*, against ciprofloxacin exposure time. **b** PCoA plot, derived from a subset of the data shown in Fig. 2b, of the two isolates at 4xMIC-20h with 95%-confident ellipses. **c** The ten most important features (vertical axis) for random forest model, built from 1000 decision trees, to distinguish SL1344 from SL1344*gyrA* with corresponding importance index (horizontal axis). **d** Radar chart for z-score transferred value of the most important features. Each line represents a datapoint of either SL1334 (blue) or SL1334*gyrA* (orange). **e** Representative images of SL1344 and SL1344*gyrA* at 4xMIC-20h taken from one of two biological replicates. Source data are provided as a Source Data file.

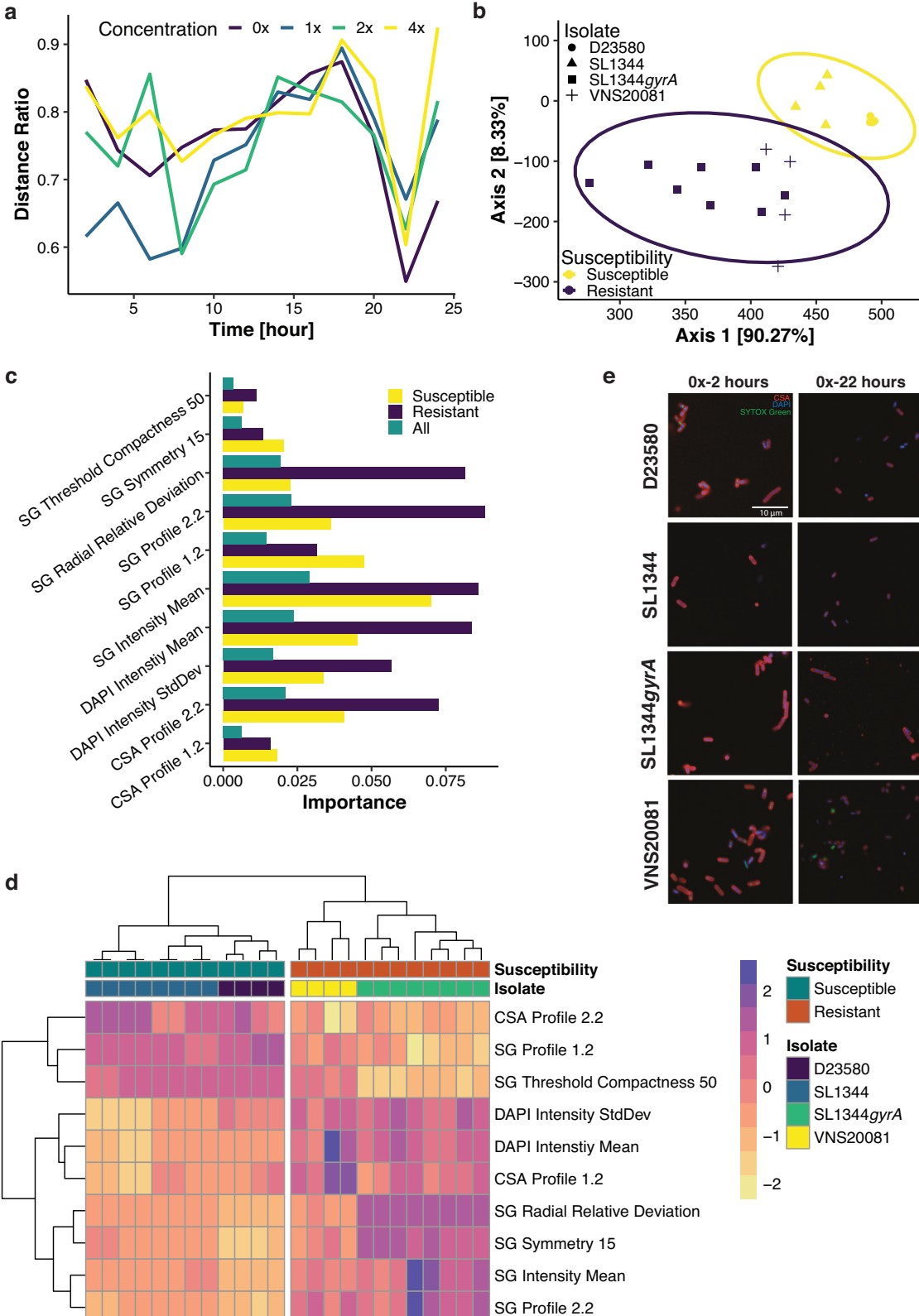

**Fig. 4 | Discrimination of resistant and susceptible isolates at 0xMIC-22h.**
**a** Distance ratio (vertical axis), of data grouped by antimicrobial susceptibility, at different drug concentration against exposure time (horizontal axis). **b** PCoA plot for all datapoints at 0xMIC-22h. A subset from PCoA data (Fig. 2b) at 0xMIC-22h was reused for this plot. Yellow (susceptible) and dark purple (resistant) ellipses are 95%-confident ellipses. **c** The ten most important features derived from random forest, with 1,000 decision trees, identifying resistant isolate at 0xMIC-22h. Vertical axis is the important features against the importance on horizontal axis. **d** Heatmap

of the ten most important features with hierarchical clustering on horizontal axis of datapoints and vertical axis of features. Each cell in the heatmap is the z-score value of a feature of a datapoint. A subset of z-score data from previous heatmap (Fig. 2a) was reused for the plot. **e** Representative images of D23580, SL1344, SL1344*gyrA* and VNS20081 at 0xMIC-2h (left column) and 0xMIC-22h (right column) taken from one well of one of two biological replicates. Source data are provided as a Source Data file.

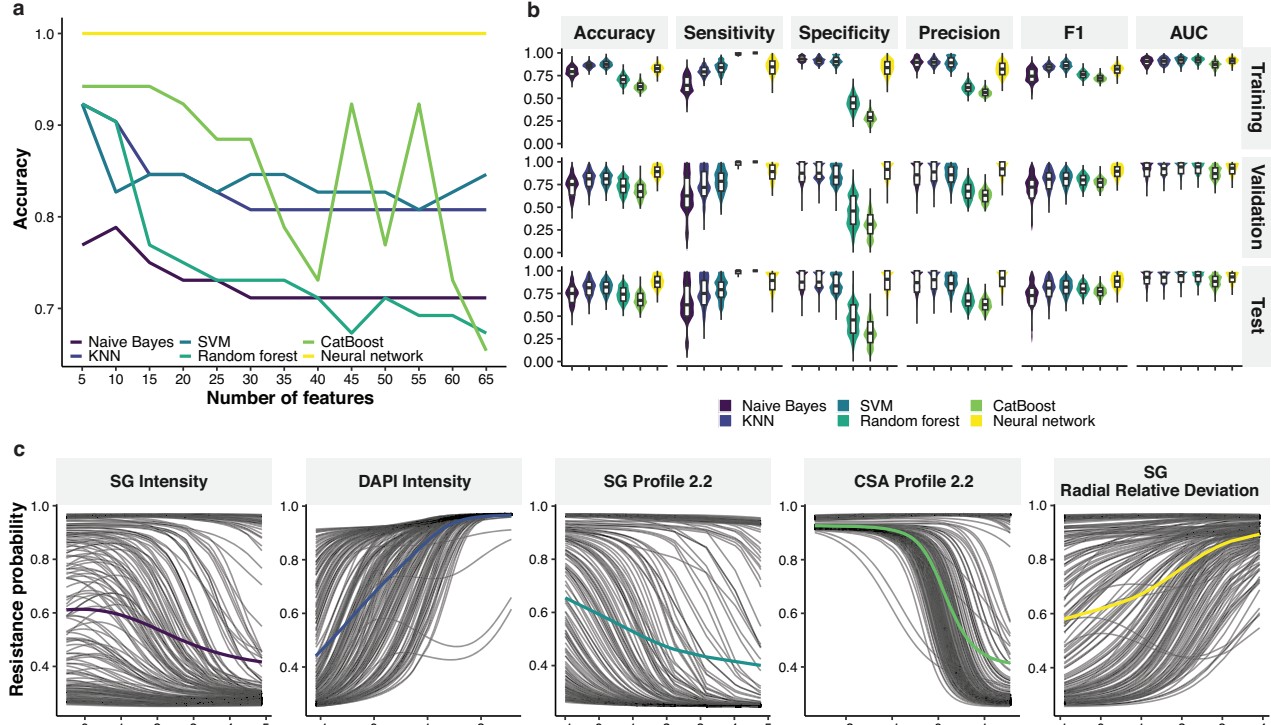

**Fig. 5 | Machine learning classifiers to identify resistant isolates at 0xMIC-22h.**
**a** Predictive performance of machine learning models set against number of most important features. Horizontal axis is number of features ranked by importance and vertical axis is accuracy on validation set. **b** Boxplots and violin plots for predictive performance of machine learning models on $n = 1000$ random training/validation/test partitions. Vertical axis is the values of performance metrics. For each boxplot, the central bar represents median of a metric, the box represents 25% (Q1) and 75% (Q3) quartile, the lower whisker represents Q1 – 1.5xIQR (interquartile range), and the upper whisker represents Q3 + 1.5xIQR. **c** Partial Dependence Plot of the five most important features (color lines) against resistance probability calculated by the neural network. Gray lines are Individual Conditional Expectation plot of each datapoint, which shows responses of the resistance probability as the feature values change. Horizontal axis is z-score of the features and vertical axis is corresponding resistance probability. Source data are provided as a Source Data file.

isolates (Fig. S4a). Therefore, the feature-selection interpretation from the model was associated with high confidence; the ten most relevant important features were measurements of fluorescence intensity and texture (Fig. 4c). These important imaging features were plotted on a heatmap with hierarchical clustering (Fig. 4d). We again found that ciprofloxacin resistant *S*. Typhimurium intrinsically segregated from susceptible organisms and had distinctive imaging data patterns (Fig. S4b, Fig. 4e), suggesting that the clustered structure of the data was driven principally by these key features. These data indicate that ciprofloxacin-susceptible and -resistant isolates can be distinguished based on a small number of imaging features without ciprofloxacin exposure.

## Machine learning classifiers can distinguish between ciprofloxacin susceptible and resistant isolates without ciprofloxacin exposure

To generalize the distinction between ciprofloxacin-susceptible and resistant isolates using an extended dataset covering most of the common ciprofloxacin resistance determinants, we generated new imaging data of 13 additional clinical strains and three of the four prior *S*. Typhimurium strains at 0 × MIC-22h (Table 1). Data from these 16 strains were combined and analysed (Fig. S5). All important features of these strains were ranked in descending order based on their relative contribution from the random forest model. The ranked features were then input stepwise into machine learning classifiers in an increment of five features to identify the best classifiers for these data. The classifiers included the Naïve Bayes classifier, K-nearest neighbour classifier, support vector machine (SVM), random forest, gradient boosting of decision trees (CatBoost), and artificial neural networks. Incorporating

many features in the machine learning classifiers diminished the predictive performance of those models, and the accuracy of most of the classifiers declined as features were added. Ultimately, we found that only five morphological features were required to reliably distinguish antimicrobial susceptibility (Fig. 5a).

To compare predictive performance among the classifiers, the machine learning models were trained with the five most important features, and performance metrics were recorded accordingly on 1,000 randomly split training, validation, and testing sets (Fig. 5b). The metrics consisted of accuracy, sensitivity, specificity, precision, F1 score, and area under the receiver operating characteristic curve (AUC) (Table S1). Among all experimental classifiers, the neural network exhibited the highest performance metrics compared to other classifiers. Specifically, on the testing sets, the neural network exhibited an average accuracy of $0.87 \pm 0.08$ (mean ± standard deviation), sensitivity of $0.87 \pm 0.11$, specificity of $0.89 \pm 0.12$, precision of $0.90 \pm 0.1$, F1 score of $0.87 \pm 0.08$, and AUC of $0.91 \pm 0.07$ (Fig. 5b).

We then sought to interpret the causal relationship between input features and the output of resistance probability from the neural network classifier (Fig. 5c). Given imaging features of an isolate, the neural network classifier predicted the corresponding resistance probability, with a higher probability indicating a greater likelihood of an isolate being resistance to ciprofloxacin. Overall, the complex neural network suggested that resistance probability responded to the important features in a nonlinear fashion, which is difficult to model with traditional statistics. Notably, the signal from SYTOX Green stain comprised 60% of the important features. As SG, a DNA stain, cannot penetrate intact cells, SG intensity is associated with bacterial cell permeability. Specifically, low SG intensity, indicating membrane

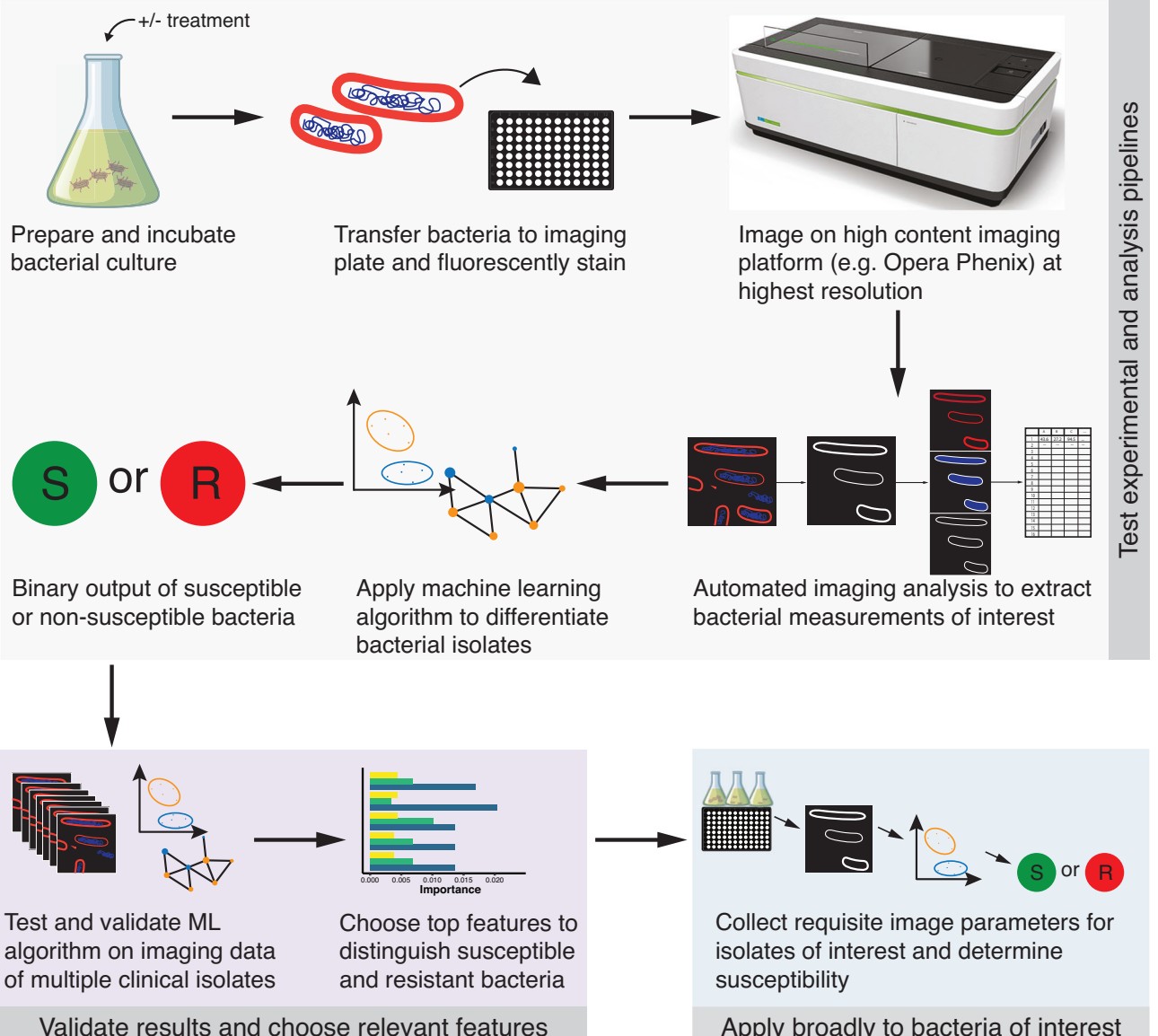

**Fig. 6 | Actualized and proposed workflow for bacterial imaging and AMR analysis.** Flowchart of the steps used to test bacteria for antimicrobial resistance. The first set of steps (steps within the gray box) describes the workflow of the experimental setup of bacterial isolates, imaging, and machine learning analysis pipeline testing. The second set of steps (purple box) indicate the training and validation phase of the method to check accuracy and identify key differentiating imaging features. The first two portions of this workflow were executed in this study. The third step (blue box) is the intended application of this method: to apply the imaging and analysis process to other bacterial isolates of interest to predict antimicrobial susceptibility. The Opera Phenix icon (top right) was created by BioRender.com, released under a Creative Commons Attribution-NonCommercial-NoDerivs 4.0 International license https://creativecommons.org/licenses/by-nc-nd/4.0/deed.en. The Erlenmeyer flask icon was produced by brgfx on Freepik.

integrity, was associated with high resistance probability (Fig. 5c). This finding suggests that even without ciprofloxacin exposure, resistant isolates might have inherently different membrane permeability than susceptible isolates.

In contrast to SG intensity, low DAPI intensity, indicating bacterial DNA content, correlated with low resistance probability, which may be explained by the membrane integrity and hypothetically by competitive DNA binding with SG (Fig. 5c). The neural network showed that high SG profile and CSA profile were associated with a low probability of resistance and high SG radial relative deviation correlated with a high probability of resistance. Overall, classification using the neural network on the additional *S.* Typhimurium strains of differing ciprofloxacin susceptibilities confirmed that ciprofloxacin-resistant *S.* Typhimurium strains are morphologically distinct from the ciprofloxacin-susceptible strains, and this distinction can be identified

by assessing specific imaging features without any prior antimicrobial exposure or knowledge of their susceptibility. Based on our optimized methods and results using the combination of HCI and machine learning, we developed a generalized workflow that could be used across a broader set of bacterial isolates (Fig. 6). This novel method can thus be tested and trained across more diverse organisms to extend its applicability.

## Discussion

AMR is a critical public health problem, and novel methodologies for screening and predicting AMR that more rapidly determine AST and drug pharmacokinetics/pharmacodynamics (PK/PD) are urgently needed. The ability to predict bacterial behaviour from morphology could have a wide-ranging set of applications beyond AMR detection in clinical microbiology and drug discovery, particularly for diagnostics

and vaccine target identification. Here, we used HCI to determine morphological differences between isolates of *S*. Typhimurium exposed to differing inhibitory concentrations of ciprofloxacin and then used machine learning models to classify the observed phenotypes. The proposed machine learning classifiers can infer the inter-relation between morphological data and the ciprofloxacin-treatment conditions which were combinations of different ciprofloxacin concentrations and different exposure times. We found that ciprofloxacin-resistant and -susceptible isolates have distinct imaging-based characteristics, independent of ciprofloxacin exposure, which can be distinguished using machine learning classifiers. To our knowledge, this is a unique example of machine learning techniques being used to predict antimicrobial-resistant and -susceptible bacterial isolates using high-content imaging data without exposure to antimicrobials.

Compared to many other methods to study AMR[35–37], HCI analysis is high-throughput and can be performed using small culture volumes to provide a high-resolution view of bacterial populations at the scale of an individual organism. Our study has shown that HCI does not require prior knowledge of resistance mechanisms or phenotypic susceptibility by traditional AST to identify resistant isolates. This differs from current high-throughput and rapid AST technologies, including the Vitek2 and the Sysmex P-100 AST platform, which rely on bacterial growth in the presence of antimicrobial and lack single-cell resolution. This makes HCI a potentially valuable AST and PK/PD determination technology. Given that other Gram-negative rod-shaped bacteria undergo similar phenotypic changes in response to ciprofloxacin and antimicrobials of other classes[13,33,38], it may be possible to extrapolate our findings to infer the responses of other Gram-negative bacteria to ciprofloxacin and other antimicrobials. For that reason, an imaging database at the family level may be sufficient to train machine learning models to predict resistance of an infection caused by AMR organisms, without labour-intensive efforts requiring the development of databases for individual species or subspecies.

The majority of contemporary studies have coupled HCI with dimensional analyses to distinguish antimicrobials with different MoA[13,15,16,18] and pathogenic isolates with different AMR profiles[33]. The disadvantage of such methods is that they do not allow quantitative prediction of new queried data points, especially ones that lie in the middle of pre-defined clusters. For example, dimensional analysis is unable to predict the resistance profile of an isolate with corresponding imaging datapoint falling between resistant and susceptible clusters. However, very few studies have employed machine learning classifiers to quantitatively predict antimicrobial MoA[14,17] and target proteins[38], and machine learning prediction of AMR without exposure to drugs is non-existent. Due to their high complexity, machine learning, especially nonlinear, models are considered to be 'black-box' tools lacking interpretability, which might prevent scientific understanding of the results[39]. In this study, we designed and implemented various 'white-box' machine learning classifiers to identify AMR isolates of *S*. Typhimurium and interpreted nonlinear relations between morphological features and AMR probability. The interpretable machine learning analysis proposed in this study could be a model for other HCI analyses, promoting more widespread use of machine learning in the field of biology, particularly clinical microbiology.

This study has limitations. While we created an HCI analysis pipeline suitable for single strain bacterial cultures, it likely needs further refinement to optimally segment bacterial cells, and our use of a proprietary analysis software may have limited the parameter flexibility. However, similar initial image segmentation and analysis can be performed using state-of-the-art deep learning architecture, including U-net[40] and Autoencoder[41]. An improved image analysis algorithm specialized for bacterial imaging data may provide improved differentiation between resistant and susceptible organisms at the single-cell level. Our study is also limited by experimental sampling methods in which averaged morphological data was extracted from images acquired on cross-sectional samples of *S*. Typhimurium populations at discrete time points. As a result, it was not possible to measure the longitudinal transformation of specific bacterial morphologies, which reduces the single-cell resolution of the assay. Furthermore, pure bacterial cultures were grown in liquid medium in a laboratory setting, which arguably makes the study less clinically relevant, as we do not know how accurately our classifier would work on polymicrobial infections or suboptimal laboratory conditions. However, this approach facilitates further studies on *S*. Typhimurium at the molecular and cellular levels, which could be leveraged for rapid AST. Additionally, our study exclusively focused on *S*. Typhimurium and ciprofloxacin because ciprofloxacin is a first-line treatment for invasive *Salmonella* infection[42,43] not investigating inter-species variability and drug-specific morphological changes that may be important for drug resistance. It is possible that this methodology may require refinement for other bug-drug combinations. However, prior studies have shown the similarities in morphology between different Gram-negative bacterial species subjected to antimicrobials[13–16,18], suggesting that these trends may be generalizable across Gram-negative bacteria. Additionally, while we were aware of the efflux pumps and porins found in the strains we used, we did not investigate their relative contributions to resistance, which may be important factors. Lastly, we recognize that this system in its current form would be prohibitively complex and expensive in most clinical laboratory systems; however, we anticipate that novel diagnostic devices could leverage the small number of important imaging features identified by this method for accurate AMR prediction.

Using various machine learning algorithms, we have conceived an analysis pipeline for interpreting and predicting ciprofloxacin resistance phenotypes of *S*. Typhimurium based on high-content confocal imaging data, which can serve as a framework for further studies. This framework can be generalized to bridge the gap between molecular information and bacterial cellular responses, spanning from novel compound or antibody targets to predicting unknown protein functions to novel resistance mechanisms. In the field of diagnostics, this could be exploited to develop high-accuracy machine-learning classifiers for rapid AST[44,45] that do not rely on traditional clinical microbiology techniques. Moreover, this framework, coupled with increasingly accessible microfluidic technology[46], could facilitate high-throughput diagnoses, thus streamlining clinical lab workflows. Although the clinical application may still be distant, this imaging and machine learning-based approach may be an important tool in the future for rapidly and accurately assessing drug resistance in the hospital context. Whilst innovative rapid AST approaches have advanced recently[47], the development of novel antimicrobials has declined in the last three decades[48]. This bleak outlook demands novel technologies to speed up the development of antimicrobial therapies. Particularly, HCI can be used to predict the MoA of new antimicrobial agents[14] providing data-driven guidance for further characterization of the agents. The technology has been employed to screen monoclonal antibodies against multi-drug resistant bacteria[20], which opens an opportunity to assess the efficiency of antibodies binding to heterogeneous populations of bacteria at individual scale. While living organisms are driven by complex molecular mechanisms, our current understanding of biological (particularly microbial) systems has been limited by how humans observe and interpret experimental data. Beyond the application of clinical microbiology, HCI aided by ML algorithms could help observe and explain nuanced cellular morphology, and, combined with genomic data, could vastly increase our understanding of the genotype-phenotype linkage. HCI, integrated with other high-throughput and automation technologies, is expected to produce massive multi-modality data. These can be combined with interpretable machine learning to generate novel hypotheses and provide biological insights to address AMR and other global health needs.

## Methods

### Ethical approval

Ethical approval for the isolates from the Democratic Republic of the Congo study was granted by the Institutional Review Board of ITM (ref. 613/08), the Ethics Committee of Antwerp University (ref. 8/20/96), and the School of Public Health of Kinshasa in DRC (ref. 074/2017). Isolates were shared via Material Transfer Agreements with ITM, INRB, and IVI, and shipments were performed according to IATA norms. Ethical approval for the surveillance study in Ghana from which isolates were obtained was granted by the IVI Institutional Review Board (IRB) and the local research ethics committee. Ethical approval for the isolates from Vietnam was provided by the scientific and ethics committees of the collaborating institutions and the Oxford Tropical Research Ethics Committee.

### Bacterial isolates and growth conditions

Prior to experimentation, all isolates (Table 1) were grown on Isosensitest agar (Oxoid, CM0471) and subjected to ciprofloxacin M.I.C.E. (Oxoid, MA0104F) or Etest (BioMerieux, 412311) to determine baseline ciprofloxacin susceptibility. Isolates were maintained on Isosensitest agar and streaked fresh weekly from frozen stocks. To prepare for imaging experiments, isolates were always inoculated into 10 ml Isosensitest broth (Oxoid, CM0473) from plates, followed by overnight shaking at 37 °C for 16–18 h.

### GyrA spontaneous mutants

To isolate spontaneous nalidixic acid mutant lines from *S.* Typhimurium isolates SL1344 and D23580, bacterial cultures were grown overnight in L-broth, and 100 μl of this was spread onto L-agar containing 100 μg/ml nalidixic acid for initial spontaneous mutant generation. After overnight incubation at 37 °C, single colonies that had grown were re-plated on L-agar also containing 100 μg/ml nalidixic acid. Any colonies that were present on these agar plates were then streaked serially onto agar plates harbouring increasing concentrations of nalidixic acid up to 400 μg/ml, then these were switched to plates containing ciprofloxacin, harbouring from 0.1 μg/ml ciprofloxacin up to 1.0 μg/ml ciprofloxacin. Once colonies were able to grow stably on 1.0 μg/ml ciprofloxacin, pure cultures were established and saved as frozen stocks. From the frozen stocks, overnight cultures were grown for genomic DNA purification and were purified using the Promega Wizard DNA Purification Kit (Promega, A1120). Following purification, DNA was PCR-amplified to check for single nucleotide polymorphisms (SNP) in the QRDR of *gyrA* using primers obtained from a prior study and manufactured by IDT: 5′-GAGATGGCCTGAAGC-3′ for nucleotides 108 to 127 and 5′-TACCGTCATAGTTATCCA CG -3′ for nucleotides 435 to 454, forward and reverse, respectively[49]. A C→A SNP change was found = in *gyrA*. To evaluate genetic differences between parent and isogenic strains, the isogenic and parent strains were grown for 24 h in Isosensitest broth prior to genomic DNA isolation for whole genome sequencing (detailed below).

### Ciprofloxacin susceptibility testing by MIC eTest

Isolates were streaked from frozen stocks on Isosensitest plates and grown at 37 °C. Three serial streaks on fresh plates were subsequently performed. For M.I.C.E. or Etest application, a few colonies from each plate were inoculated in ~3 ml PBS and vortexed well to create a slightly cloudy solution. 100 μl of the solution was spotted on Isosensitest plates and spread well before gently laying down the MIC test strip. Inoculated and control plates were incubated overnight at 37 °C and then visually analysed. Each *S.* Typhimurium isolate was tested a minimum of two times to ensure an accurate reading.

### Time kill curves

Four *S.* Typhimurium isolates were chosen for the initial time kill curve analysis, performed as in Sridhar et al.[32]. These were D23580, SL1344,

SL1344*gyrA*, VNS20081[50–53]. Initially, colonies from plates were inoculated into 10 ml of Isosensitest broth, and these were shaken at 200 rpm at 37 °C overnight. 10 μl of the subsequent culture was then added to 990 μl of 1x PBS to make a 1:100 dilution for the inoculum. 100 μl of this preparation was added to 10 ml of Isosensitest containing different levels (0x, 1x, 2x, 4x MIC) of ciprofloxacin according to the predetermined MIC of each isolate (μl). The starter inoculum was between 1 and $5 \times 10^5$ CFU/ml. Cultures were incubated shaking at 37 °C, and aliquots were taken to determine colony forming units (CFU) at 0, 2, 4, 6, 8, and 24 h. For this analysis, serial dilutions were made using samples of each culture, and a total of 50 μl of each dilution was plated using 10 μl spots of inoculum onto L-agar. CFUs were counted and determined as CFU/ml. Means and standard deviations (SD) of three replicates per isolate were calculated.

### Bacterial whole genome sequencing

Library preparation for Illumina sequencing was undertaken at the Wellcome Sanger Institute using automated systems using the IHTP WGS NEB Ultra II library kit. Libraries were sequenced on an Illumina HiSeq platform (Illumina, San Diego, USA) using standard running protocols. Illumina adapter content was removed from the reads using Trimmomatic v.0.33. Reads mapping was undertaken using the WSI bacterial mapping pipeline, which uses bwa, and de novo assembly was performed using Velvet[54]. For SL1344*gyrA* and D23580*gyrA* mutants, Illumina HiSeq reads for the isogenic mutants and parental strains were mapped to the parental reference strain: SL1344 (FQ312003.1) and D23580 (FN424405.1), respectively, using SMALT v0.7.4 (sanger.-ac.uk/resources/software/smalt/) to produce a BAM file[51,55]. Briefly, variant detection was performed as detailed here: SAMtools mpileup v0.1.19 with parameters -d 1000 -DSugBf and bcftools v0.1.19[56] were used to generate a BCF file of all variant sites. The bcftools variant quality score was set as greater than 50, mapping quality was set as greater than 30, the allele frequency was determined as either 0 for bases called same as the reference or 1 for bases called as a SNP (af1 < 0.95), the majority base call was set to be present in at least 75% of reads mapping at the base (ratio < 0.75), the minimum mapping depth was four reads, a minimum of two of the four had to map to each strand, and strand_bias was set as less than 0.001, map bias less than 0.001, and tail_bias less than 0.001. Bases that did not meet those criteria were called uncertain and removed. A pseudogenome was constructed by substituting the base calls in the BCF file in the reference genome. Recombinant regions in the chromosome, such as prophage regions, were removed from the alignment and checked using Gubbins v1.4.10[57]. SNP sites were extracted from the alignment using snp-sites and analysed manually. SNPs in *gyrA* identified by PCR were confirmed. For SL1344*gyrA*, a SNP (C→A) at position 2373805 was found to confer a D87Y mutation in GyrA. Isolate accession numbers are in Table S2.

### Opera Phenix imaging

Two separate Opera Phenix experiments were performed. The first experiment was a 24 h evaluation of bacterial growth under four ciprofloxacin concentrations (0×, 1×, 2×, 4×MIC) at two-hour increments (2, 4, 6, 8, 10, 12, 14, 16, 18, 20, 22, 24) conducted on four isolates SL1344, SL1344*gyrA*, D23580, and VNS20081, which has been previously described in ref. 32. 150 μl of each isolate was inoculated 1:1000 in 150 ml Isosensitest broth with each appropriate concentration of ciprofloxacin in a 200 ml flask and incubated at 37 °C shaking at 200 rpm. Following 2 h of growth, 10 ml of each culture was removed from the flask, and the flask was returned to the incubator. The 10 ml fraction was centrifuged at 3200 x *g* for 7 min at 4 °C, and the supernatant was decanted. The pellet was resuspended in 100 μl PBS, and 50 μl was added to two wells of a vitronectin-coated Opera CellCarrier Ultra-96 well plate (Perkin Elmer, 6055302). The plate was statically incubated at 37 °C for 10 min, after which the bacterial cultures were

aspirated, fixed in 4% paraformaldehyde (PFA) for 10 min, and washed with 1x PBS. After fixation, the plate of fixed bacteria was kept at 4 °C until the next time point. The same protocol was followed for each time point with this exception: once there was sufficient bacterial growth (as assessed visually), 10 ml of bacteria was still removed from the flask at each time point; however, the culture was either centrifuged and resuspended in 250 μl or 50 μl dense cultures were added neat to wells. An average of $5 \times 10^6$ bacteria was added to each well. Upon completion of the 24 h period, wells were incubated with 2% bovine serum albumin (BSA) for 30 min and then for 1 h at ambient temperature in the dark with CSA-Alexa Fluor 647 (Novus Biologicals, NB110-16952AF647) at 1:1000 in BSA. Wells were aspirated and then incubated with solutions containing 1:100 4′,6-diamidino-2-phenylindole (DAPI) (Invitrogen, D1306) and 1:200 SYTOX Green (Invitrogen, S7020) for 20 min. Wells were washed 1× with PBS; plates were sealed and imaged. Imaging was performed using the 63x water immersion objective in confocal mode on an Opera Phenix high content imaging platform. For each well, 40 fields across two planes (distance of 0.5 nm) were evaluated. Number of bacteria imaged per well varied by experiment and isolate, but the average per well was between 600 and 2000, and the lowest number of bacteria captured in any well across all isolates and replicates was 80. Three biological replicates of this experiment were performed for D23580 and VNS20081; two biological replicates were performed for SL1344 and SL1344gyrA. Two technical replicate wells were used for each condition in every biological replicate.

The second Opera Phenix imaging experiment used 16 of the 17 isolates detailed in Table 1, including all except D23580. Here, the only time point assessed was 22 h, and no ciprofloxacin treatment was used. However, to maintain experimental consistency with the previous experiment, cultures were grown like before in 150 ml Isosensitest, and 10 ml of culture was removed every two hours to mimic the change in growth condition. At the 22 h time point, 50 μl of each culture was added (neat) to two wells of a vitronectin-coated Opera CellCarrier Ultra-96 well plate, and the same fixation and staining protocol as above was used.

## Opera Phenix analysis

Analysis was performed using a Perkin Elmer Harmony software analysis pipeline designed for *S.* Typhimurium, as previously described[13,32]. Briefly, inputted images were subjected to flatfield correction, and images were calculated using the DAPI and CSA (AlexaFluor 647) channels. Image calculations were refined by size and shape characteristics. A linear classifier was applied to the filtered population, single bacteria were identified, and morphology and intensity characteristics were calculated Feature names outputted were in some cases specific to the Harmony software (e.g. "SG Profile 2.2".) The output of the Harmony analysis was tabulated by plate and object, and results were further analysed and visualised in R (v 3.6.1)[58] using packages 'dplyr' and 'ggplot2'[59]. Adobe Illustrator was used to format images and graphs for presentation.

## AMR in silico analysis

ARIBA (v2.14.6)[60] using the Comprehensive Antimicrobial Resistance Database (CARD, v3.1.3)[61] with default parameters was used on reads data of all isolates to determine AMR genes. Results were cross-checked using ResFinder[62].

## Machine Learning data analysis

Averaged morphological data was analysed using Matlab (R2021a), Python (version 3.5), and R (version 4.0.5)[58] programming language based on toolbox, package, and library availability. PCoA was performed using 'vegan' package[63]. Heatmap and spider plots were generated with 'pheatmap'[64] and 'fmsb'[65] packages respectively. Other plots were created with 'ggplot2'[59] and 'ggsci'[66] packages in R. Pairwise

Kruskal-Wallis tests were performed using 'ggpubr' package[67]. For feature selection, 'randomForestSRC' package[68] was utilized to train a random forest model with default parameters and 1,000 decision trees. Features were selected and ranked decreasingly by importance index[69] extracted from the random forests.

To assess segregation of the isolates, we defined an index named distance ratio $r(X_1, \ldots, X_k, \ldots, X_n)$ as the following equation

$$r(X_1, \ldots, X_k, \ldots, X_n) = \frac{\mathbb{E}\left(\|\boldsymbol{x}^{(i)} - \boldsymbol{x}^{(j)}\|_2\right)\boldsymbol{x}^{(i)}\epsilon X_k, \boldsymbol{x}^{(j)}\epsilon X_k}{\mathbb{E}\left(\|\boldsymbol{x}^{(i)} - \boldsymbol{x}^{(j)}\|_2\right)\boldsymbol{x}^{(i)}\epsilon X_k, \boldsymbol{x}^{(j)}\notin X_k} \quad (1)$$

where $X_k$ is data group, e.g., resistant versus susceptible, $\boldsymbol{x}^{(i)}$ is a datapoint, $\mathbb{E}(.)$ is statistical mean, and $\|.\|_2$ is Euclidean norm. A low distance ratio indicates data points within one group are closely related while well separated from ones in other groups.

The 'Statistics and Machine learning' toolbox on Matlab was used to train the Naïve Bayes classifier, KNN classifier, SVM, and random forest. Neural network and CatBoost were trained with the 'Neural network' toolbox on Matlab and 'CatBoost' library[70] on Python, respectively. Data were split randomly into training, validation and test sets with a ratio of 50:25:25. Training sets were used to estimate parameters of the machine learning models, while validation sets were used to choose hyperparameters. Finally, models' performance was calculated on the test sets. Hyperparameters are the configurations to control the learning process and cannot be estimated from data. Grid-search algorithm was employed to optimize the hyperparameters of all the machine-learning models. Optimal hyperparameters were chosen, on the validation set, corresponding to the highest accuracy defined as

$$ACC = \frac{TP + TN}{P + N} \quad (2)$$

where P was the number of actual resistant isolates in the data, N was the number of actual susceptible isolates, TP was the number of accurately predicted resistant isolates, and TN was the number of accurately predicted susceptible isolates. Other metrics for performance evaluation, including sensitivity (SEN), specificity (SPE, precision (PRE), and F1 score, can be described as the follows

$$SEN = \frac{TP}{P} \quad (3)$$

$$SPE = \frac{TN}{N} \quad (4)$$

$$PRE = \frac{TP}{PP} \quad (5)$$

$$F1 = \frac{2PRE \times SEN}{PRE + SEN} \quad (6)$$

where PP was number of predicted resistant isolates. AUC was calculated with 'perfcurve' function on Matlab. For interpretation of the machine learning models, particularly neural networks, an in-house code was built to generate Partial Dependence Plot (PDP)[71] and Individual Conditional Expectation (ICE)[72] plot. In the ICE plot, for each datapoint in $\{(\boldsymbol{x}_S^{(i)}, \boldsymbol{x}_C^{(i)})\}_{i=1}^N$, the curve $\widehat{f_S^{(i)}}$ representing models' output is plotted against $\boldsymbol{x}_S^{(i)}$, while $\boldsymbol{x}_C^{(i)}$ remains fixed, where N is the size of the dataset, i is index of the ith datapoint, $\boldsymbol{x}_S$ is the interpreted feature, $x_C$ is other input features. PDP was defined as average of ICE

$$\widehat{f_{\boldsymbol{x}_S}}(\boldsymbol{x}_S) = \mathbb{E}_{\boldsymbol{x}_S}\left[\widehat{f}(\boldsymbol{x}_S, \boldsymbol{x}_C)\right] = \int \widehat{f}(\boldsymbol{x}_S, \boldsymbol{x}_C) d\mathbb{P}(\boldsymbol{x}_C) \quad (7)$$

In practice, the partial function $\widehat{f_{x_s}}$ is estimated by calculating averages on the dataset

$$\widehat{f_{x_s}}(x_S) = \frac{1}{N}\sum_{i=1}^{N}\widehat{f}\left(x_S, x_C^{(i)}\right) \qquad (8)$$

**Statistics and Reproducibility.** No statistical method was used to predetermine sample size. No data were excluded from the analyses. The experiments were not randomized. The investigators were not blinded to allocation during experiments and outcome assessment.

## Reporting summary

Further information on research design is available in the Nature Portfolio Reporting Summary linked to this article.

## Data availability

High-content imaging data will be provided upon request. Source data used to generate figures are provided with this paper as file "Source Data.xlsx". Isolate accession numbers are listed in Supplementary Table 2. Source data are provided with this paper.

## Code availability

Code used in this study is available at https://github.com/Tuan-AnhTran/Cip_STM (https://doi.org/10.5281/zenodo.11004491).

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

## Acknowledgements

We are grateful for the assistance of Ursula Panzer and Liselotte Hardy for managing and organizing isolates. We also thank Sally Forrest and Josefin Bartholdson-Scott for assistance in Opera Phenix image and analysis pipeline optimization. This project was supported by Wellcome senior research fellowship to 215515/Z/19/Z (SB). Purchase of the Opera Phenix was supported by an Innovate UK Commercial in Confidence grant. SS was funded through a Wellcome Trust studentship (206194). TAT was funded through an Oxford-Oak scholarship. The funders had no role in the design and conduct of the study; collection, management, analysis, and interpretation of the data; preparation, review, or approval of the manuscript; and decision to submit the manuscript for publication.

## Author contributions

T.A.T., S.S., S.R., and S.B. conceived the study. S.S. and S.B. performed laboratory investigations. T.A.T., S.S., B.T.N., P.T.B. conducted data analysis. Isolates and isolate data were provided by O.L., J.J., S.V.P., and F.M. The original draft was prepared by T.A.T. and S.S., revised by T.A.T., S.S., and S.B., and then reviewed and edited by all authors. Supervision was provided by S.B., N.R.T., G.D., and P.T.B.

## Competing interests

The authors declare no competing interests.

## Additional information

[1]The Department of Medicine, University of Cambridge, Cambridge, UK. [2]Oxford University Clinical Research Unit, Ho Chi Minh City, Vietnam. [3]Nuffield Department of Medicine, University of Oxford, Oxford, UK. [4]The Wellcome Sanger Institute, Hinxton, Cambridge, UK. [5]Sanofi, Kymab, Babraham Research Campus, Cambridge, UK. [6]Department of Microbiology, Institut National de Recherche Biomédicale, Kinshasa, Democratic Republic of Congo. [7]Service de Microbiologie, Cliniques Universitaires de Kinshasa, Kinshasa, Democratic Republic of Congo. [8]Department of Microbiology, Immunology and Transplantation, KU Leuven, Leuven, Belgium. [9]Department of Clinical Sciences, Institute of Tropical Medicine, Antwerp, Belgium. [10]Laboratory of Medical Microbiology, Vaccine and Infectious Disease Institute, University of Antwerp, Antwerp, Belgium. [11]International Vaccine Institute, 1 Gwanak-ro, Gwanak-gu, Seoul 08826, Republic of Korea. [12]Heidelberg Institute of Global Health, University of Heidelberg, Heidelberg, Germany. [13]Madagascar Institute for Vaccine Research, University of Antananarivo, Antananarivo, Madagascar. [14]London School of Hygiene and Tropical Medicine, London, UK. [15]Faculty of Mathematics and Computer Science, University of Science, Vietnam National University Ho Chi Minh City, Ho Chi Minh City, Vietnam. [16]Information Science Faculty, Saigon University, Ho Chi Minh City, Vietnam. [17]IAVI, Chelsea and Westminster Hospital, London, UK. [18]These authors contributed equally: Tuan-Anh Tran, Sushmita Sridhar. ✉e-mail: sgb47@cam.ac.uk

