## [Peer Review File · Nature Communications]

REVIEWER COMMENTS

Reviewer #1 (Remarks to the Author):

The manuscript entitled “Combining machine learning with high-content imaging to infer ciprofloxacin susceptibility in clinical isolates of *Salmonella Typhimurium*” by Tran and Sridhar et al. describes a novel approach utilizing high content imaging paired with machine learning to predict ciprofloxacin susceptibility in *Salmonella Typhimurium* isolates. The authors begin by demonstrating that there are quantifiable differences in four different *Salmonella Typhimurium* strains exposed to ciprofloxacin. Using a random forest classifier, the authors determine which morphology properties are most important for distinguishing ciprofloxacin concentrations, treatment times, and strains. Next, the authors demonstrate that there are morphological differences between resistant and susceptible strains without exposing these bacteria to ciprofloxacin. These morphological properties are defined, refined, and utilized by several different machine learning classifiers to discern ciprofloxacin susceptibility of 13 additional clinical isolates.

Overall, the manuscript was rigorous and tells an interesting story.

The major strength of this manuscript is the demonstration that high content imaging and machine learning can be used to predict ciprofloxacin susceptibility without the need for drug exposure. This capability is novel, would save time and resources compared to traditional AST, and has significant potential utility for the field of antimicrobial resistance. An additional strength is the demonstration of this technology with multiple clinical isolates from different regions.

The major disadvantages of this manuscript include the uncertainty of how well this method would work for different antibiotics, different/mixed bacterial species, and diverse resistance mechanisms. However, the majority of these limitations were discussed, and some points are perhaps outside the scope of the current manuscript to address experimentally. The other major weaknesses are the lack of a well-defined workflow and the presentation of figures (see below).

General concerns:

Many of the figures need to be re-formatted for consistency and so that they are large enough to read. All font sizes for axis, keys, etc., should be the same size across figures.

Scale bars are missing from HCl images.

“*gyrA*” is italicized in the text, but not in the figures.

Please ensure figures are colorblind friendly.

What would the ultimate readout for this assay be (as in sensitive vs resistant, or more of a gradient that would require interpretation)? What would the potential workflow look like? The authors demonstrated proof-of-principal, but do not explicitly define a fully realized workflow for how this technology could be utilized by others. A flowchart might help address this.

Page 9 lines 225-229 and page 14 lines 356-357: The authors hypothesize that DNA gyrase mutations were responsible for the morphology differences between sensitive and resistant isolates. The authors

also point out that porins and efflux pumps can drive resistance. However, all of the resistant and intermediate strains/isolates used here had a mutation of some kind in *gyrA*. Clinically, how common is it common to find resistant bacteria without mutations in *gyrA*? It would improve the significance of this study if the authors could demonstrate that susceptibility can be predicted in bacteria where resistance is driven by something other than mutations in *gyrA*.

The following are minor concerns, which predominately relate to improving the clarity of the manuscript:

Results:

Page 4 lines 92-94: It would be helpful if the authors would briefly explain the mechanism of action of ciprofloxacin.

Page 5 lines 104-106: The use of two clinical isolates and two laboratory strains increases the rigor of this study. However, can the authors briefly define the what makes VNS20081 and SL1344*gyrA* resistant (what is *gyrA*)?

Page 5 lines 123-124: It would be helpful if the authors explained up front what stains/antibodies were chosen for HCl and why, instead of waiting until page 11.

Page 7 lines 166-167, page 9 lines 206-215 and page 10 lines 234-240: The authors state that the most important features used to train the random forest classifier are given in Figure 2d, Figure 3c-d, and Figure 4c-d. Why are there different numbers after parameters like symmetry and profile? If the naming of these parameters is specific to a particular software, then stating this would help clarify this for the reader. The brief explanation of the type of parameters (lines 170-174) was extremely helpful and it would be worthwhile to do the same for Figures 3c-d and 4c-d.

Page 8 lines 179-181: Figure S2a should be referenced.

Page 8 lines 182-183: Please be sure to specify that this trend in Figure 2e does not include SL1344*gyrA*.

Page 8 lines 188-190: The authors state that bacterial cell length becomes similar over time for D23580, VNS20081, and SL1344 at all treatment concentrations and reference Figure S2b-c. However, the images shown in Figure S2b at 4x MIC do not support this statement. Please consider rephrasing.

Page 10 lines 248-249: This subtitle seems to fit the previous section more aptly. Wasn't the neural network chosen as the best classifier?

Page 11 line 270: ROC needs to be defined, or should not be abbreviated since it is only used twice in the text.

Pages 11-12 lines 281-283: The statement "...low FITC intensity, indicating membrane integrity, was associated with high resistance probability (Figure 5c). This finding suggests that under exposure to

ciprofloxacin, resistant isolates were more likely to survive and to [retain an] intact membrane, while susceptible isolates died, leading to higher membrane permeability..." suggests that these bacteria were exposed to ciprofloxacin. It doesn't appear that drug treatment was used here based on the materials and methods, but the previous statement makes this unclear. If bacteria were untreated, can the authors speculate on another explanation for why low FITC correlated with increased resistance probability?

Discussion:

Page 12 lines 298-300: Somewhere in the discussion, can the authors elaborate on how this technology could be applied to other areas of biology besides clinical microbiology?

Page 13 line 324: Check the formatting of the references 2, 3, 4, 5, 7.

Page 14 lines 356-357: Can the authors speculate on how well this method would be able to detect resistance driven by new, less common, and diverse resistance-mechanisms? Would traditional AST methods be more reliable in these instances? Since ciprofloxacin interferes with DNA replication, the use of DNA binding stains makes sense, but would addition of other stains be needed to broaden the application of this technology to other antibiotics?

Materials and Methods:

Page 19 line 471: A few more details would help make this method more versatile for other imaging platforms. What objective was used (20x, 40x, or does it matter)? Was there a minimum number of bacteria imaged per well? What is the optimal cell density for a well? What is the average imaging time for a 96-well or 384-well plate?

Page 22 lines 535-537: Can the authors elaborate on what this training entails?

Page 22 line 538: What are "hyperparameters"?

Figures:

Figure 1a: It would be easier to compare treated versus untreated if these data points were combined on one graph. Could treated be represented with dotted lines and untreated be represented with solid lines (or some variation)?

Figure 1c: Who is "SDS012"? I am guessing blue was supposed to be "D23580"?

Figure 1: It would help make comparison easier within figures and between all figures if the authors keep the order (and color) of bacterial strains consistent (D23580, VNS20081, SL1334, SL1334gyrA for example).

Figure 2b: Even zoomed in on a computer screen, this figure is very hard to see. If this was due to limited space, figure 2c could be moved to the supplemental.

Figure 3: Can the authors move the images from Figure S3b, along with representative images for each bacteria at 0xMIC-0h to Figure 3? This clearly demonstrates the differences in morphology between the

two strains.

Figure 3c and 4c: It unclear why there are 3 “importance index” values for each parameter. What is “importance”? What is “all” referring to? Why are there different numbers for SL1334 and SL1334gyrA? A brief description in the text and legend would be helpful. Also, since “SYTOX Green” was used, it may be more straightforward if “SYTOX Green” is written instead of “FITC” for all pertinent figures.

Figure 4: Representative images of bacteria at 0xMIC-0h and 0xMIC-22h would help demonstrate the difference in morphology of these strains.

Figure 4d: Are there multiple replicates for each strain/isolate (4 replicates for D23580 and VNS20081; 8 replicates for SL1334 and SL1334gyrA)? If so, what is considered a replicate? If not, why are there multiple pink, purple, teal, and blue boxes for each respective strain/isolate?

Figure 5b: The authors should explain what “training”, “validation”, and “test” sets mean. What is different between these populations?

Reviewer #2 (Remarks to the Author):

This study is very limited and only focuses on two laboratory-typed strains and two clinical *S. Typhimurium* isolates. Only four isolates is too tiny a dataset to make any concrete conclusions. Moreover, *S. Typhimurium* is not a clinically relevant pathogen like the ESKAPE pathogens.

The authors are making some really big claims, like predicting drug susceptibility of clinical bacterial isolates, by using this very small and non-relevant dataset. For any potential clinical microbiological relevance, these studies should be done.

Also, different bacteria have different features, including different strains of the same species. The significant limitation of the study is that the amount of training data is relatively low, which raises concerns about overfitting. The model may be overfitted and have limited transferability. The robustness of the method will depend on the specific microscope setups, cameras, strains used, etc.

Also, infection is often mixed with multiple different strains of bacteria. Thus, it is important to be able to differentiate a mixed pool of bacteria by the proposed method.

Most importantly bacterial morphological characteristics are affected by the growth environments. It is important that analysis in mixed bacterial samples and in infected tissues or biological fluids rather than isolates grown in rich culture media would provide the reliability of the described approach.

It also seems like the authors are not aware of the area of AST testing. Contrary to the authors, there are several automated and highthroughput AST testing methods like BD Phoenix and Vitek2.

The use of microscopy and ML in AMR and AST is not new. There does exist a commercial tool using confocal microscopy for AST, sysmex.

It is a very costly method to use of confocal microscope to identify drug-resistant bacteria more and in understanding the phenotypic impact of antimicrobials on the bacterial cell in order to identify drugs with new modes of action. Most microbiology labs dont have these.

The authors should provide the loss graphs. Also, the image database, codes, and trained network should become publicly available to validate the results presented. The authors will have to provide more convincing data to fully demonstrate the performance of this method.

Reviewer #1 (Remarks to the Author):

The manuscript entitled “Combining machine learning with high-content imaging to infer ciprofloxacin susceptibility in clinical isolates of Salmonella Typhimurium” by Tran and Sridhar et al. describes a novel approach utilizing high content imaging paired with machine learning to predict ciprofloxacin susceptibility in Salmonella Typhimurium isolates. The authors begin by demonstrating that there are quantifiable differences in four different Salmonella Typhimurium strains exposed to ciprofloxacin. Using a random forest classifier, the authors determine which morphology properties are most important for distinguishing ciprofloxacin concentrations, treatment times, and strains. Next, the authors demonstrate that there are morphological differences between resistant and susceptible strains without exposing these bacteria to ciprofloxacin. These morphological properties are defined, refined, and utilized by several different machine learning classifiers to discern ciprofloxacin susceptibility of 13 additional clinical isolates.

Overall, the manuscript was rigorous and tells an interesting story. The major strength of this manuscript is the demonstration that high content imaging and machine learning can be used to predict ciprofloxacin susceptibility without the need for drug exposure. This capability is novel, would save time and resources compared to traditional AST, and has significant potential utility for the field of antimicrobial resistance. An additional strength is the demonstration of this technology with multiple clinical isolates from different regions.

The major disadvantages of this manuscript include the uncertainty of how well this method would work for different antibiotics, different/mixed bacterial species, and diverse resistance mechanisms. However, the majority of these limitations were discussed, and some points are perhaps outside the scope of the current manuscript to address experimentally. The other major weaknesses are the lack of a well-defined workflow and the presentation of figures (see below).

We appreciate this thoughtful feedback from the reviewer. We have attempted to address these concerns in our Discussion, particularly in the limitations section, have improved the figure and workflow presentation, and have separately addressed each individual comment below.

General concerns:

Many of the figures need to be re-formatted for consistency and so that they are large enough to read. All font sizes for axis, keys, etc., should be the same size across figures.

We thank the reviewer for this guidance and have accordingly re-sized the figures and labels for consistency.

Scale bars are missing from HCI images.

These have been added to all image panels.

“gyrA” is italicized in the text, but not in the figures.

We have italicized *gyrA* in all figures.

Please ensure figures are colorblind friendly.

Thank you for this suggestion. We have modified the colors in plots to use colorblind-friendly palettes.

What would the ultimate readout for this assay be (as in sensitive vs resistant, or more of a gradient that would require interpretation)? What would the potential workflow look like? The authors demonstrated proof-of-principle, but do not explicitly define a fully realized workflow for how this technology could be utilized by others. A flowchart might help address this.

We appreciate this valuable feedback and have addressed this by including a generalized flowchart for the workflow (Figure 6), lines 304-308. In our case, the ultimate readout was ciprofloxacin susceptibility versus non-susceptibility (binary output), and that could similarly be applied to other antimicrobials. but depending on the user's need, the image analysis pipelines and subsequent machine learning algorithms could likely be modulated.

Page 9 lines 225-229 and page 14 lines 356-357: The authors hypothesize that DNA gyrase mutations were responsible for the morphology differences between sensitive and resistant isolates. The authors also point out that porins and efflux pumps can drive resistance. However, all of the resistant and intermediate strains/isolates used here had a mutation of some kind in *gyrA*. Clinically, how common is it common to find resistant bacteria without mutations in *gyrA*? It would improve the significance of this study if the authors could demonstrate that susceptibility can be predicted in bacteria where resistance is driven by something other than mutations in *gyrA*.

*We thank the reviewer for the suggestion and point about *gyrA* versus other mechanisms of ciprofloxacin resistance. However, we note that in our final tested dataset, we included an isolate (5390_4) that only had a *qnrS* gene and isolate 8314_12 that carried a *GyrB* not *GyrA* mutation. The general accuracy of our algorithm determining the correct susceptibility of the test set was very high.*

*Moreover, in *Salmonella Typhimurium*, the main driver for ciprofloxacin resistance is a mutation in *GyrA*. While studies (e.g. Cuypers et al, 2018; Chang et al, 2021) have noted that isolates may have only efflux pumps or *qnrS/qnrB* genes, the general understanding is that these mechanisms amplify resistance but typically are not sufficient to make a susceptible isolate resistant and are often carried in isolates with *GyrA* mutations. We have highlighted this point in the text by indicating at lines 110-113 that mutations in *GyrA* are the predominant mechanisms of ciprofloxacin resistance in the two non-susceptible isolates of *Salmonella Typhimurium* that we used.*

The following are minor concerns, which predominately relate to improving the clarity of the manuscript:

Results:

Page 4 lines 92-94: It would be helpful if the authors would briefly explain the mechanism of action of ciprofloxacin.

We have added a phrase to describe ciprofloxacin's effect on replication in lines 97-98 and have added references 23-25 to guide readers.

Page 5 lines 104-106: The use of two clinical isolates and two laboratory strains increases the rigor of this study. However, can the authors briefly define the what makes VNS20081 and SL1344gyrA resistant (what is gyrA)?

We have addressed this in lines 110-112 to explain that VNS20081 and SL1344gyrA both have mutations in the quinolone resistance determining region of GyrA.

Page 5 lines 123-124: It would be helpful if the authors explained up front what stains/antibodies were chosen for HCl and why, instead of waiting until page 11.

We appreciate this suggestion and have added a sentence describing the stains used in lines 130-132.

Page 7 lines 166-167, page 9 lines 206-215 and page 10 lines 234-240: The authors state that the most important features used to train the random forest classifier are given in Figure 2d, Figure 3c-d, and Figure 4c-d. Why are there different numbers after parameters like symmetry and profile? If the naming of these parameters is specific to a particular software, then stating this would help clarify this for the reader. The brief explanation of the type of parameters (lines 170-174) was extremely helpful and it would be worthwhile to do the same for Figures 3c-d and 4c-d.

This nomenclature is specific to the software used, and these parameter numbers refer to distinct measurements of cellular and imaging features. As a result, we do not have specific insight into what each number means. We have addressed this in the text by stating in the Methods that these parameter names are software-specific (line 542).

Page 8 lines 179-181: Figure S2a should be referenced.

Figure S2a was referenced in line 187.

Page 8 lines 182-183: Please be sure to specify that this trend in Figure 2e does not include SL1344gyrA.

We have added language in the text at line 192 to specify that this trend does not include SL1344gyrA.

Page 8 lines 188-190: The authors state that bacterial cell length becomes similar over time for D23580, VNS20081, and SL1344 at all treatment concentrations and reference Figure S2b-c. However, the images shown in Figure S2b at 4x MIC do not support this statement. Please consider rephrasing.

Thank you for this valuable point. We have revised the language in line 200 to indicate that D23580 at 4x MIC was an exception to this finding.

Page 10 lines 248-249: This subtitle seems to fit the previous section more aptly. Wasn't the neural network chosen as the best classifier?

Thank you for this important point. We have changed the subtitle (line 258) to reflect that we compared multiple machine learning algorithms.

Page 11 line 270: ROC needs to be defined, or should not be abbreviated since it is only used twice in the text.

We have removed reference to ROC to increase clarity for the reader in lines 280 (Results) and 586 (Methods).

Pages 11-12 lines 281-283: The statement "...low FITC intensity, indicating membrane integrity, was associated with high resistance probability (Figure 5c). This finding suggests that under exposure to ciprofloxacin, resistant isolates were more likely to survive and to [retain an] intact membrane, while susceptible isolates died, leading to higher membrane permeability..." suggests that these bacteria were exposed to ciprofloxacin. It doesn't appear that drug treatment was used here based on the materials and methods, but the previous statement makes this unclear. If bacteria were untreated, can the authors speculate on another explanation for why low FITC correlated with increased resistance probability?

The reviewer has raised a very good point, and we appreciate the critical reading of the text. Based on our analysis, we found that there were inherent differences in the morphology characteristics of resistant and susceptible isolates. Our best hypothesis for this result of difference in untreated isolates is that the membranes of susceptible isolates are more permeable to SYTOX Green.

We have addressed this in the text, lines 291-293, and have also removed the confusing phrasing regarding ciprofloxacin exposure of these isolates.

Discussion:

Page 12 lines 298-300: Somewhere in the discussion, can the authors elaborate on how this technology could be applied to other areas of biology besides clinical microbiology?

We refer to the applications of drug discovery and monoclonal antibody efficacy screening in lines 396-400. However, to broaden the application further, we have included in lines 402-404 the potential for HCl + machine learning technology to elucidate genotype-phenotype linkages through the ability to screen collections of organisms and evaluate genotypic differences.

Page 13 line 324: Check the formatting of the references 2, 3, 4, 5, 7.

We have corrected this formatting error.

Page 14 lines 356-357: Can the authors speculate on how well this method would be able to detect resistance driven by new, less common, and diverse resistance-mechanisms? Would traditional AST methods be more reliable in these instances? Since ciprofloxacin interferes with DNA replication, the use of DNA binding stains makes sense, but would addition of other stains be needed to broaden the application of this technology to other antibiotics?

Because we only tested the case of ciprofloxacin susceptibility in Salmonella Typhimurium, we cannot confidently predict the accuracy of this method for new resistance mechanisms. However, given the magnitude of image data captured that feeds into the machine learning algorithm, we believe that this approach would have high accuracy for other resistance mechanisms, particularly in cases where there is a clear morphological change due to drug treatment. Regarding the second question about stains, we believe that the combination of stains used (membrane, nucleic acid, live/dead) may be an optimal set of stains to capture a variety of features needed to analyze bacterial morphological changes. However, for given bacteria and drug combinations, it may be better to use alternative stain combinations.

To address these limitations, as it was beyond the scope of this study, we have added a line (373) in the limitations section of our Discussion to acknowledge the need for further testing.

Materials and Methods:

Page 19 line 471: A few more details would help make this method more versatile for other imaging platforms. What objective was used (20x, 40x, or does it matter?)? Was there a minimum number of bacteria imaged per well? What is the optimal cell density for a well? What is the average imaging time for a 96-well or 384-well plate?

We have accordingly added details about the bacterial density and imaging platform used in lines 513, 518-522. Specifically, we have included information regarding the objective, fields imaged per well, and minimum number of bacteria counted.

Page 22 lines 535-537: Can the authors elaborate on what this training entails?

This has been addressed in lines 570-589

Page 22 line 538: What are “hyperparameters”?

We have added the definition of hyperparameters in the manuscript, lines 575-576.

Figures:

Figure 1a: It would be easier to compare treated versus untreated if these data points were combined on one graph. Could treated be represented with dotted lines and untreated be represented with solid lines (or some variation)?

We tried the suggested layout, but the plot was overcrowded and became difficult to distinguish treated and untreated isolates. For better clarification, we have retained the split plots in treated and untreated panels. Given that the two panels are plotted on the same scale, we believe it is easier to compare the growth dynamics at the two conditions.

Figure 1c: Who is “SDS012”? I am guessing blue was supposed to be “D23580”?

We have fixed this in Figure 1c. Thank you for noticing this error.

Figure 1: It would help make comparison easier within figures and between all figures if the authors keep the order (and color) of bacterial strains consistent (D23580, VNS20081, SL1334, SL1334gyrA for example).

These have been addressed in the figures.

Figure 2b: Even zoomed in on a computer screen, this figure is very hard to see. If this was due to limited space, figure 2c could be moved to the supplemental.

We have done this by increasing the size of the complex plots and text. Thank you.

Figure 3: Can the authors move the images from Figure S3b, along with representative images for each bacterium at 0xMIC-0h to Figure 3? This clearly demonstrates the differences in morphology between the two strains.

Thank you for this suggestion. We have done this and accordingly updated the figure legend for Figure 3 and references to Figures 3 and S3 (lines 220, 222).

Figure 3c and 4c: It unclear why there are 3 “importance index” values for each parameter. What is “importance”? What is “all” referring to? Why are there different numbers for SL1334 and SL1334gyrA? A brief description in the text and legend would be helpful. Also, since “SYTOX Green” was used, it may be more straightforward if “SYTOX Green” is written instead of “FITC” for all pertinent figures.

Importance or importance index is a metric to measure how important a feature is for the machine learning model to make prediction accurately. If an important feature is “removed” (e.g. through permutation) from the dataset, the model performance will sustainably drop. We have referenced citation number 68 in line 560 to define the importance index.

The random forest model can report the importance regarding each classification class, e.g. susceptible vs resistant or SL1344 vs SL1344gyrA. “All” means the important index for each feature regarding all the classes.

We have changed the confusion of SYTOX Green versus FITC in the text and figures to just use SYTOX Green in all cases.

Figure 4: Representative images of bacteria at 0xMIC-0h and 0xMIC-22h would help demonstrate the difference in morphology of these strains.

Thank you for this suggestion. We did not conduct imaging at 0 hours, but we have added panels for 0xMIC-2h and 0xMIC-22h as Figure 4e to represent the difference between the earliest point captured and 22 hours. We have referenced Figure 4e in lines 251-252.

Figure 4d: Are there multiple replicates for each strain/isolate (4 replicates for D23580 and VNS20081; 8 replicates for SL1334 and SL1334gyrA)? If so, what is considered a replicate? If not, why are there multiple pink, purple, teal, and blue boxes for each respective strain/isolate?

The data used in figure 4d was generated across two biological replicates with two technical replicate wells for each isolate. In the case of SL1344 and SL1344gyrA, two planes of the plate were imaged due to uneven cell adherence. For D23580 and VNS20081, only one plane was captured, as cell adherence was high. Data was aggregated per well and per plane. We have clarified the imaging techniques used in the Materials and Methods section.

Figure 5b: The authors should explain what “training”, “validation”, and “test” sets mean. What is different between these populations?

We have added the definitions of training, validation and test sets in lines 572-575.

Reviewer #2 (Remarks to the Author):

This study is very limited and only focuses on two laboratory-typed strains and two clinical S. Typhimurium isolates. Only four isolates is too tiny a dataset to make any concrete conclusions. Moreover, S. Typhimurium is not a clinically relevant pathogen like the ESKAPE pathogens.

We thank the reviewer for this feedback. While we acknowledge that this study was limited in scope, this was intended as a proof-of-concept to illustrate that the combination of high-content imaging data and machine learning analysis can yield important insights into resistant and susceptible organisms. While there were only three distinct isolates used, these are genomically different and the two clinical isolates were deliberately chosen as they belong to clinically important sequence types and are found in distinct geographic and clinical contexts. Additionally, in order to validate our findings from the four strains tested, we performed extensive analysis on an additional 13 clinical isolates with diverse resistance profiles (as elaborated in the Results section and Figure 5). These results validated our initial findings, as the machine learning algorithm was able to accurately predict the susceptibility of the additional isolates.

Furthermore, we contend that Salmonella Typhimurium is a clinically relevant pathogen in parts of the world and warrants attention. It is estimated that invasive non-typhoidal Salmonella causes 681,000 deaths annually globally. In particular, non-typhoidal Salmonella is considered one of the most important bacterial pathogens in sub-Saharan Africa and is a leading cause of invasive infection and mortality in children under five

years. Moreover, ciprofloxacin resistance is a growing problem in Salmonella Typhimurium, and this exacerbates the risk for mortality, highlighting the importance of better understanding and addressing ciprofloxacin-resistant Salmonella, as we have done in this study.

The authors are making some really big claims, like predicting drug susceptibility of clinical bacterial isolates, by using this very small and non-relevant dataset. For any potential clinical microbiological relevance, these studies should be done.

We appreciate the critique of our findings; however, we have explained the clinical relevance of ciprofloxacin-resistant Salmonella Typhimurium in our Introduction and highlighted the limitations of our study in the Discussion. We feel confident in our findings, as they apply to ciprofloxacin resistance in Salmonella, and as discussed in our Discussion, we think there is broader applicability to other Gram-negative organisms, based on morphology data and analysis that has been performed in other studies. These studies have found that the morphological changes in Gram-negative bacteria due to antimicrobial exposure are conserved, which suggests that we would see similar imaging signatures if this imaging and machine learning method were applied to other Gram-negative organisms. We have added a further sentence in the limitations section of our Discussion to acknowledge that adaptations to the imaging and machine learning pipelines may be required for other bacteria. However, we would like to reiterate that this study was not intended as a final product ready for deployment in a clinical laboratory; this was a proof-of-concept study, in which we were able to distinguish resistant from susceptible Salmonella using imaging data.

Also, different bacteria have different features, including different strains of the same species. The significant limitation of the study is that the amount of training data is relatively low, which raises concerns about overfitting. The model may be overfitted and have limited transferability. The robustness of the method will depend on the specific microscope setups, cameras, strains used, etc.

We were aware of the small number of instances in our dataset. Therefore, we trained and validated the machine learning models on 1,000 random splits of training/validation/test sets with a ratio of 50:25:25 to ensure the vigorousness of performance estimation for each model. Indeed, the average accuracy of the best model (neural network) on training sets was 0.83 ± 0.06 and that on test sets was 0.87 ± 0.08 , suggesting that overfitting was less likely and the model was even slightly underfitted. We expect that the predictive performance will increase with more data.

With regards to the specific experimental setup, we have specified that such image capture and analysis is intended for high content imaging systems. These instruments have highly calibrated cameras embedded to facilitate single bacterial cell resolution image capture. However, we would like to point out that the isolates tested in the validation section of this study were predominantly clinical isolates, and therefore represent "wild type" organisms that have not been extensively passaged in the laboratory and thus reflect the variation that one might expect from other clinical isolates. Additionally, we refer the reviewer to references 13 and 15, which

describes optimization of high content imaging screening across multiple Gram-negative and Gram-positive organisms.

Also, infection is often mixed with multiple different strains of bacteria. Thus, it is important to be able to differentiate a mixed pool of bacteria by the proposed method.

We stated in the limitations within our Discussion that we had not tested this on mixed bacterial populations (lines 367-368). The objective of our study was not to differentiate between bacterial species in a mixed infection, thus we believe that this point is outside the scope of our research. Additionally, even gold standard methods of antimicrobial susceptibility testing used in clinical laboratories first obtain single colonies from blood or other sample types before testing for resistance. In the case of polymicrobial infections, morphologically distinct colonies are independently streaked out and carried forward for testing. Thus, while we fully agree with the reviewer that there is a clinical need for methods that identify the constituents and resistance profiles of bacteria within a mixed infection, this was not the goal of our study. However, given the ability of our method to distinguish between resistant and susceptible isolates within a species, we believe that it would be possible to train an algorithm to distinguish between bacterial species within a given sample.

Most importantly bacterial morphological characteristics are affected by the growth environments. It is important that analysis in mixed bacterial samples and in infected tissues or biological fluids rather than isolates grown in rich culture media would provide the reliability of the described approach.

We appreciate this comment on the importance of physiological relevant experimental conditions. While this is a valuable point, the purpose of our study was to test a method and provide proof that it could be reproduced reliably. To do so, we used standard laboratory conditions for bacterial growth. All bacterial isolates were streaked out on Isosensitest agar and cultured in Isosensitest broth, which is one of the two standard growth media used for antimicrobial susceptibility testing, as it is better defined and less rich than Luria-Burtani broth (the default medium used in research settings). Thus, we believe that our growth media closely follows the conditions used in a clinical laboratory setting. While it is true that we did not directly test isolates in biological fluids, we would like to point out that doing so would reduce the reproducibility of this study, and this is also not how cultures are tested in clinical laboratories.

It also seems like the authors are not aware of the area of AST testing. Contrary to the authors, there are several automated and highthroughput AST testing methods like BD Phoenix and Vitek2.

We thank the reviewer for raising this topic. We have added text in lines 68-70 of the Introduction and 329-332 of the Discussion to address the existence of high-throughput AST technologies and how they differ from our proposed method. However, the Vitek2 and BD Phoenix, while semi-automated methods, still use traditional AST phenotyping methods and readouts. We contend that the novelty of our developed method is that it can predict antimicrobial susceptibility without exposure to the antimicrobial. We

acknowledge that our proposed method requires further optimization for deployment in a clinical laboratory context. However, we believe that the novelty of data captured and method of analysis make it unique and could lead to downstream diagnostic products and workflows that improve diagnostic AST capability.

The use of microscopy and ML in AMR and AST is not new. There does exist a commercial tool using con focal microscopy for AST, sysmex.

Thank you for this comment. The Sysmex platform (PA-100 AST System) is a microfluidic-device with a phase-contrast microscope specifically designed for urinary samples with current application for five bacterial species. While this method does leverage microscopy, it performs by detecting bacterial growth or lack of growth in the presence of antimicrobial on a panel. In this regard, it uses the same principles as traditional phenotypic AST. We appreciate the mention of this technology and have added reference to it in our Discussion in lines 329-332. However, we believe that this platform provides limited information on bacterial susceptibility, as the resolution of image data collected is low and is not comprehensively analyzed.

It is a very costly method to use of confocal microscope to identify drug-resistant bacteria more and in understanding the phenotypic impact of antimicrobials on the bacterial cell in order to identify drugs with new modes of action. Most microbiology labs dont have these.

Thank you for this important consideration regarding cost. Indeed, we agree that high-content imaging would be prohibitively expensive for use in most microbiology laboratories. However, one of our key findings was that a machine learning algorithm was able to distinguish with high accuracy susceptible bacteria using only five imaging features. We believe that a lower cost diagnostic could be designed around the capture and analysis of a limited set of imaging features, and we hope that an outcome of our study is that groups that focus on diagnostic development can use our results to trial such methods. We have addressed this in lines 378-381 of the Discussion.

The authors should provide the loss graphs. Also, the image database, codes, and trained network should become publicly available to validate the results presented. The authors will have to provide more convincing data to fully demonstrate the performance of this method.

We appreciate the reviewer's suggestions regarding the report of model performance. We understand that loss curve is a useful technique to diagnose the iterative training process of a machine model, such as a neural network. However, not all machine learning models in this study were trained in an iterative manner, e.g. the Naïve Bayes classifier. Additionally, we trained and validated the machine learning models on 1,000 random splits, which would generate 1,000 distinct loss curves for each model (assuming that all models can generate loss curves). For that reason, we decided not to include the loss curves in our analysis. Instead, we reported accuracy, sensitivity, specificity, precision, F1 score, and AUC on training, validation, and test sets of 1,000 random splits. These metrics are standard demonstration of performance of machine learning models, which is well-accepted by the machine learning community. We also employed Partial Dependence Plot and Individual Conditional Expectation to

demonstrate the interpretability of our methods. Finally, raw data and code to train the models are available on a public github repository.